# Learning Bound for Parameter Transfer Learning

**Wataru Kumagai**
Faculty of Engineering
Kanagawa University
kumagai@kanagawa-u.ac.jp

## Abstract

We consider a transfer-learning problem by using the parameter transfer approach, where a suitable parameter of feature mapping is learned through one task and applied to another objective task. Then, we introduce the notion of the local stability and parameter transfer learnability of parametric feature mapping, and thereby derive a learning bound for parameter transfer algorithms. As an application of parameter transfer learning, we discuss the performance of sparse coding in self-taught learning. Although self-taught learning algorithms with plentiful unlabeled data often show excellent empirical performance, their theoretical analysis has not been studied. In this paper, we also provide the first theoretical learning bound for self-taught learning.

## 1 Introduction

In traditional machine learning, it is assumed that data are identically drawn from a single distribution. However, this assumption does not always hold in real-world applications. Therefore, it would be significant to develop methods capable of incorporating samples drawn from different distributions. In this case, *transfer learning* provides a general way to accommodate these situations. In transfer learning, besides the availability of relatively few samples related with an objective task, abundant samples in other domains that are not necessarily drawn from an identical distribution, are available. Then, transfer learning aims at extracting some useful knowledge from data in other domains and applying the knowledge to improve the performance of the objective task. In accordance with the kind of knowledge that is transferred, approaches to solving transfer-learning problems can be classified into cases such as instance transfer, feature representation transfer, and parameter transfer (Pan and Yang (2010)). In this paper, we consider the *parameter transfer* approach, where some kind of parametric model is supposed and the transferred knowledge is encoded into parameters. Since the parameter transfer approach typically requires many samples to accurately learn a suitable parameter, unsupervised methods are often utilized for the learning process. In particular, transfer learning from unlabeled data for predictive tasks is known as *self-taught learning* (Raina et al. (2007)), where a joint generative model is not assumed to underlie unlabeled samples even though the unlabeled samples should be indicative of a structure that would subsequently be helpful in predicting tasks. In recent years, self-taught learning has been intensively studied, encouraged by the development of strong unsupervised methods. Furthermore, sparsity-based methods such as sparse coding or sparse neural networks have often been used in empirical studies of self-taught learning.

Although many algorithms based on the parameter transfer approach have empirically demonstrated impressive performance in self-taught learning, some fundamental problems remain. First, the theoretical aspects of the parameter transfer approach have not been studied, and in particular, no learning bound was obtained. Second, although it is believed that a large amount of unlabeled data help to improve the performance of the objective task in self-taught learning, it has not been sufficiently clarified how many samples are required. Third, although sparsity-based methods are typically employed in self-taught learning, it is unknown how the sparsity works to guarantee the performance of self-taught learning.

The aim of the research presented in this paper is to shed light on the above problems. We first consider a general model of parametric feature mapping in the parameter transfer approach. Then, we newly formulate the local stability of parametric feature mapping and the parameter transfer learnability for this mapping, and provide a theoretical learning bound for parameter transfer learning algorithms based on the notions. Next, we consider the stability of sparse coding. Then we discuss the parameter transfer learnability by dictionary learning under the sparse model. Applying the learning bound for parameter transfer learning algorithms, we provide a learning bound of the sparse coding algorithm in self-taught learning.

This paper is organized as follows. In the remainder of this section, we refer to some related studies. In Section 2, we formulate the stability and the parameter transfer learnability of the parametric feature mapping. Then, we present a learning bound for parameter transfer learning. In Section 3, we show the stability of the sparse coding under perturbation of the dictionaries. Then, by imposing sparsity assumptions on samples and by considering dictionary learning, we derive the parameter transfer learnability for sparse coding. In particular, a learning bound is obtained for sparse coding in the setting of self-taught learning. In Section 4, we conclude the paper.

### 1.1 Related Works

Approaches to transfer learning can be classified into some cases based on the kind of knowledge being transferred (Pan and Yang (2010)). In this paper, we consider the parameter transfer approach. This approach can be applied to various notable algorithms such as sparse coding, multiple kernel learning, and deep learning since the dictionary, weights on kernels, and weights on the neural network are regarded as parameters, respectively. Then, those parameters are typically trained or tuned on samples that are not necessarily drawn from a target region. In the parameter transfer setting, a number of samples in the source region are often needed to accurately estimate the parameter to be transferred. Thus, it is desirable to be able to use unlabeled samples in the source region.

Self-taught learning corresponds to the case where only unlabeled samples are given in the source region while labeled samples are available in the target domain. In this sense, self-taught learning is compatible with the parameter transfer approach. Actually, in Raina et al. (2007) where self-taught learning was first introduced, the sparse coding-based method is employed and the parameter transfer approach is already used regarding the dictionary learnt from images as the parameter to be transferred. Although self-taught learning has been studied in various contexts (Dai et al. (2008); Lee et al. (2009); Wang et al. (2013); Zhu et al. (2013)), its theoretical aspects have not been sufficiently analyzed. One of the main results in this paper is to provide a first theoretical learning bound in self-taught learning with the parameter transfer approach. We note that our setting differs from the environment-based setting (Baxter (2000), Maurer (2009)), where a distribution on distributions on labeled samples, known as an environment, is assumed. In our formulation, the existence of the environment is not assumed and labeled data in the source region are not required.

Self-taught learning algorithms are often based on sparse coding. In the seminal paper by Raina et al. (2007), they already proposed an algorithm that learns a dictionary in the source region and transfers it to the target region. They also showed the effectiveness of the sparse coding-based method. Moreover, since remarkable progress has been made in unsupervised learning based on sparse neural networks (Coates et al. (2011), Le (2013)), unlabeled samples of the source domain in self-taught learning are often preprocessed by sparsity-based methods. Recently, a sparse coding-based generalization bound was studied (Mehta and Gray (2013); Maurer et al. (2012)) and the analysis in Section 3.1 is based on (Mehta and Gray (2013)).

## 2 Learning Bound for Parameter Transfer Learning

### 2.1 Problem Setting of Parameter Transfer Learning

We formulate parameter transfer learning in this subsection. We first briefly introduce notations and terminology in transfer learning (Pan and Yang (2010)). Let $\mathcal{X}$ and $\mathcal{Y}$ be a sample space and a label space, respectively. We refer to a pair of $\mathcal{Z} := \mathcal{X} \times \mathcal{Y}$ and a joint distribution $P(\mathbf{x}, y)$ on $\mathcal{Z}$ as a *region*. Then, a *domain* comprises a pair consisting of a sample space $\mathcal{X}$ and a marginal probability of $P(\mathbf{x})$ on $\mathcal{X}$ and a *task* consists of a pair containing a label set $\mathcal{Y}$ and a conditional distribution $P(y|\mathbf{x})$. In addition, let $\mathcal{H} = \{h : \mathcal{X} \to \mathcal{Y}\}$ be a hypothesis space and $\ell : \mathcal{Y} \times \mathcal{Y} \to \mathbb{R}_{\geq 0}$

represent a loss function. Then, the expected risk and the empirical risk are defined by $\mathcal{R}(h) := \mathbb{E}_{(\mathbf{x},y) \sim P} [\ell(y, h(\mathbf{x}))]$ and $\widehat{\mathcal{R}}_n(h) := \frac{1}{n} \sum_{j=1}^{n} \ell(y_j, h(\mathbf{x}_j))$, respectively. In the setting of transfer learning, besides samples from a region of interest known as a target region, it is assumed that samples from another region known as a source region are also available. We distinguish between the target and source regions by adding a subscript $\mathcal{T}$ or $\mathcal{S}$ to each notation introduced above, (e.g. $P_{\mathcal{T}}, \mathcal{R}_{\mathcal{S}}$). Then, the homogeneous setting (i.e., $\mathcal{X}_{\mathcal{S}} = \mathcal{X}_{\mathcal{T}}$) is not assumed in general, and thus, the heterogeneous setting (i.e., $\mathcal{X}_{\mathcal{S}} \neq \mathcal{X}_{\mathcal{T}}$) can be treated. We note that self-taught learning, which is treated in Section 3, corresponds to the case when the label space $\mathcal{Y}_{\mathcal{S}}$ in the source region is the set of a single element.

We consider the parameter transfer approach, where the knowledge to be transferred is encoded into a parameter. The parameter transfer approach aims to learn a hypothesis with low expected risk for the target task by obtaining some knowledge about an effective parameter in the source region and transfer it to the target region. In this paper, we suppose that there are parametric models on both the source and target regions and that their parameter spaces are partly shared. Then, our strategy is to learn an effective parameter in the source region and then transfer a part of the parameter to the target region. We describe the formulation in the following. In the target region, we assume that $\mathcal{Y}_{\mathcal{T}} \subset \mathbb{R}$ and there is a parametric feature mapping $\psi_{\boldsymbol{\theta}} : \mathcal{X}_{\mathcal{T}} \to \mathbb{R}^m$ on the target domain such that each hypothesis $h_{\mathcal{T},\boldsymbol{\theta},\mathbf{w}} : \mathcal{X}_{\mathcal{T}} \to \mathcal{Y}_{\mathcal{T}}$ is represented by

$$h_{\mathcal{T},\boldsymbol{\theta},\mathbf{w}}(\mathbf{x}) := \langle \mathbf{w}, \psi_{\boldsymbol{\theta}}(\mathbf{x}) \rangle \tag{1}$$

with parameters $\boldsymbol{\theta} \in \Theta$ and $\mathbf{w} \in \mathcal{W}_{\mathcal{T}}$, where $\Theta$ is a subset of a normed space with a norm $\| \cdot \|$ and $\mathcal{W}_{\mathcal{T}}$ is a subset of $\mathbb{R}^m$. Then the hypothesis set in the target region is parameterized as

$$\mathcal{H}_{\mathcal{T}} = \{ h_{\mathcal{T},\boldsymbol{\theta},\mathbf{w}} | \boldsymbol{\theta} \in \Theta, \mathbf{w} \in \mathcal{W}_{\mathcal{T}} \}.$$

In the following, we simply denote $\mathcal{R}_{\mathcal{T}}(h_{\mathcal{T},\boldsymbol{\theta},\mathbf{w}})$ and $\widehat{\mathcal{R}}_{\mathcal{T}}(h_{\mathcal{T},\boldsymbol{\theta},\mathbf{w}})$ by $\mathcal{R}_{\mathcal{T}}(\boldsymbol{\theta}, \mathbf{w})$ and $\widehat{\mathcal{R}}_{\mathcal{T}}(\boldsymbol{\theta}, \mathbf{w})$, respectively. In the source region, we suppose that there exists some kind of parametric model such as a sample distribution $P_{\mathcal{S},\boldsymbol{\theta},\mathbf{w}}$ or a hypothesis $h_{\mathcal{S},\boldsymbol{\theta},\mathbf{w}}$ with parameters $\boldsymbol{\theta} \in \Theta$ and $\mathbf{w} \in \mathcal{W}_{\mathcal{S}}$, and a part $\Theta$ of the parameter space is shared with the target region. Then, let $\boldsymbol{\theta}_{\mathcal{S}}^* \in \Theta$ and $\mathbf{w}_{\mathcal{S}}^* \in \mathcal{W}_{\mathcal{S}}$ be parameters that are supposed to be effective in the source region (e.g., the true parameter of the sample distribution, the parameter of the optimal hypothesis with respect to the expected risk $\mathcal{R}_{\mathcal{S}}$); however, explicit assumptions are not imposed on the parameters. Then, the parameter transfer algorithm treated in this paper is described as follows. Let $N$- and $n$-samples be available in the source and target regions, respectively. First, a parameter transfer algorithm outputs the estimator $\widehat{\boldsymbol{\theta}}_N \in \Theta$ of $\boldsymbol{\theta}_{\mathcal{S}}^*$ by using $N$-samples. Next, for the parameter

$$\mathbf{w}_{\mathcal{T}}^* := \underset{\mathbf{w} \in \mathcal{W}_{\mathcal{T}}}{\operatorname{argmin}} \mathcal{R}_{\mathcal{T}}(\boldsymbol{\theta}_{\mathcal{S}}^*, \mathbf{w})$$

in the target region, the algorithm outputs its estimator

$$\widehat{\mathbf{w}}_{N,n} := \underset{\mathbf{w} \in \mathcal{W}_{\mathcal{T}}}{\operatorname{argmin}} \widehat{R}_{\mathcal{T},n}(\widehat{\boldsymbol{\theta}}_N, \mathbf{w}) + \rho r(\mathbf{w})$$

by using $n$-samples, where $r(\mathbf{w})$ is a 1-strongly convex function with respect to $\| \cdot \|_2$ and $\rho > 0$. If the source region relates to the target region in some sense, the effective parameter $\boldsymbol{\theta}_{\mathcal{S}}^*$ in the source region is expected to also be useful for the target task. In the next subsection, we regard $\mathcal{R}_{\mathcal{T}}(\boldsymbol{\theta}_{\mathcal{S}}^*, \mathbf{w}_{\mathcal{T}}^*)$ as the baseline of predictive performance and derive a learning bound.

## 2.2 Learning Bound Based on Stability and Learnability

We newly introduce the local stability and the parameter transfer learnability as below. These notions are essential to derive a learning bound in Theorem 1.

**Definition 1** (Local Stability). *A parametric feature mapping $\psi_{\boldsymbol{\theta}}$ is said to be locally stable if there exist $\epsilon_{\boldsymbol{\theta}} : \mathcal{X} \to \mathbb{R}_{>0}$ for each $\boldsymbol{\theta} \in \Theta$ and $L_{\psi} > 0$ such that for $\boldsymbol{\theta}' \in \Theta$*

$$\|\boldsymbol{\theta} - \boldsymbol{\theta}'\| \leq \epsilon_{\boldsymbol{\theta}}(\mathbf{x}) \Rightarrow \|\psi_{\boldsymbol{\theta}}(\mathbf{x}) - \psi_{\boldsymbol{\theta}'}(\mathbf{x})\|_2 \leq L_{\psi}\|\boldsymbol{\theta} - \boldsymbol{\theta}'\|.$$

We term $\epsilon_{\boldsymbol{\theta}}(\mathbf{x})$ the permissible radius of perturbation for $\boldsymbol{\theta}$ at $\mathbf{x}$. For samples $\mathbf{X}^n = \{\mathbf{x}_1, \dots \mathbf{x}_n\}$, we denote as $\epsilon_{\boldsymbol{\theta}}(\mathbf{X}^n) := \min_{j \in [n]} \epsilon_{\boldsymbol{\theta}}(\mathbf{x}_j)$, where $[n] := \{1, \dots, n\}$ for a positive integer $n$. Next, we formulate the parameter transfer learnability based on the local stability.

**Definition 2** (Parameter Transfer Learnability). *Suppose that $N$-samples in the source domain and $n$-samples $\mathbf{X}^n$ in the target domain are available. Let a parametric feature mapping $\{\psi_{\boldsymbol{\theta}}\}_{\boldsymbol{\theta}\in\Theta}$ be locally stable. For $\bar{\delta} \in [0,1)$, $\{\psi_{\boldsymbol{\theta}}\}_{\boldsymbol{\theta}\in\Theta}$ is said to be* parameter transfer learnable *with probability $1 - \bar{\delta}$ if there exists an algorithm that depends only on $N$-samples in the source domain such that, the output $\widehat{\boldsymbol{\theta}}_N$ of the algorithm satisfies*

$$\Pr\left[\|\widehat{\boldsymbol{\theta}}_N - \boldsymbol{\theta}^*_{\mathcal{S}}\| \le \epsilon_{\boldsymbol{\theta}^*_{\mathcal{S}}}(\mathbf{X}^n)\right] \ge 1 - \bar{\delta}.$$

In the following, we assume that parametric feature mapping is bounded as $\|\psi_{\boldsymbol{\theta}}(\mathbf{x})\|_2 \le R_\psi$ for arbitrary $\mathbf{x} \in \mathcal{X}$ and $\boldsymbol{\theta} \in \Theta$ and linear predictors are also bounded as $\|\mathbf{w}\|_2 \le R_{\mathcal{W}}$ for any $\mathbf{w} \in \mathcal{W}$. In addition, we suppose that a loss function $\ell(\cdot, \cdot)$ is $L_\ell$-Lipschitz and convex with respect to the second variable. We denote as $R_r := \sup_{\mathbf{w}\in\mathcal{W}} |r(\mathbf{w})|$. Then, the following learning bound is obtained, where the strong convexity of the regularization term $\rho r(\mathbf{w})$ is essential.

**Theorem 1** (Learning Bound). *Suppose that the parametric feature mapping $\psi_{\boldsymbol{\theta}}$ is locally stable and an estimator $\widehat{\boldsymbol{\theta}}_N$ learned in the source region satisfies the parameter transfer learnability with probability $1 - \bar{\delta}$. When $\rho = L_\ell R_\psi \sqrt{\frac{8(32+\log(2/\delta))}{R_r n}}$, the following inequality holds with probability $1 - (\delta + 2\bar{\delta})$:*

$$\mathcal{R}_{\mathcal{T}}\left(\widehat{\boldsymbol{\theta}}_N, \widehat{\mathbf{w}}_{N,n}\right) - \mathcal{R}_{\mathcal{T}}(\boldsymbol{\theta}^*_{\mathcal{S}}, \mathbf{w}^*_{\mathcal{T}})$$

$$\le \; L_\ell R_\psi \left(R_{\mathcal{W}}\sqrt{2\log(2/\delta)} + 2\sqrt{2R_r(32+\log(2/\delta))}\right)\frac{1}{\sqrt{n}} + L_\ell L_\psi R_\psi \left\|\widehat{\boldsymbol{\theta}}_N - \boldsymbol{\theta}^*_{\mathcal{S}}\right\|$$

$$+ L_\ell\sqrt{L_\psi R_{\mathcal{W}} R_\psi}\left(\frac{R_r}{2(32+\log(2/\delta))}\right)^{\frac{1}{4}} n^{\frac{1}{4}}\sqrt{\left\|\widehat{\boldsymbol{\theta}}_N - \boldsymbol{\theta}^*_{\mathcal{S}}\right\|}. \tag{2}$$

If the estimation error $\|\widehat{\boldsymbol{\theta}}_N - \boldsymbol{\theta}^*_{\mathcal{S}}\|$ can be evaluated in terms of the number $N$ of samples, Theorem 1 clarifies which term is dominant, and in particular, the number of samples required in the source domain such that this number is sufficiently large compared to the samples in the target domain.

## 2.3 Proof of Learning Bound

We prove Theorem 1 in this subsection. In this proof, we omit the subscript $\mathcal{T}$ for simplicity. In addition, we denote $\boldsymbol{\theta}^*_{\mathcal{S}}$ simply by $\boldsymbol{\theta}^*$. We set as

$$\widehat{\mathbf{w}}^*_n \; := \; \operatorname*{argmin}_{\mathbf{w}\in\mathcal{W}} \frac{1}{n}\sum_{j=1}^n \ell(y_j, \langle \mathbf{w}, \psi_{\boldsymbol{\theta}^*}(\mathbf{x}_j)\rangle) + \rho r(\mathbf{w}).$$

Then, we have

$$\mathcal{R}_{\mathcal{T}}\left(\widehat{\boldsymbol{\theta}}_N, \widehat{\mathbf{w}}_{N,n}\right) - \mathcal{R}_{\mathcal{T}}(\boldsymbol{\theta}^*, \mathbf{w}^*)$$

$$= \; \mathbb{E}_{(\mathbf{x},y)\sim P}\left[\ell(y, \langle \widehat{\mathbf{w}}_{N,n}, \psi_{\widehat{\boldsymbol{\theta}}_N}(\mathbf{x})\rangle)\right] - \mathbb{E}_{(\mathbf{x},y)\sim P}\left[\ell(y, \langle \widehat{\mathbf{w}}_{N,n}, \psi_{\boldsymbol{\theta}^*}(\mathbf{x})\rangle)\right]$$

$$+ \mathbb{E}_{(\mathbf{x},y)\sim P}\left[\ell(y, \langle \widehat{\mathbf{w}}_{N,n}, \psi_{\boldsymbol{\theta}^*}(\mathbf{x})\rangle)\right] - \mathbb{E}_{(\mathbf{x},y)\sim P}\left[\ell(y, \langle \widehat{\mathbf{w}}^*_n, \psi_{\boldsymbol{\theta}^*}(\mathbf{x})\rangle)\right] \tag{3}$$

$$+ \mathbb{E}_{(\mathbf{x},y)\sim P}\left[\ell(y, \langle \widehat{\mathbf{w}}^*_n, \psi_{\boldsymbol{\theta}^*}(\mathbf{x})\rangle)\right] - \mathbb{E}_{(\mathbf{x},y)\sim P}\left[\ell(y, \langle \mathbf{w}^*, \psi_{\boldsymbol{\theta}^*}(\mathbf{x})\rangle)\right].$$

In the following, we bound three parts of (3). First, we have the following inequality with probability $1 - (\delta/2 + \bar{\delta})$:

$$\mathbb{E}_{(\mathbf{x},y)\sim P}\left[\ell(y, \langle \widehat{\mathbf{w}}_{N,n}, \psi_{\widehat{\boldsymbol{\theta}}_N}(\mathbf{x})\rangle)\right] - \mathbb{E}_{(\mathbf{x},y)\sim P}\left[\ell(y, \langle \widehat{\mathbf{w}}_{N,n}, \psi_{\boldsymbol{\theta}^*}(\mathbf{x})\rangle)\right]$$

$$\le \; L_\ell R_{\mathcal{W}}\mathbb{E}_{(\mathbf{x},y)\sim P}\left[\left\|\psi_{\widehat{\boldsymbol{\theta}}_N}(\mathbf{x}) - \psi_{\boldsymbol{\theta}^*}(\mathbf{x})\right\|\right]$$

$$\le \; L_\ell R_{\mathcal{W}}\frac{1}{n}\sum_{j=1}^n \left\|\psi_{\widehat{\boldsymbol{\theta}}_N}(\mathbf{x}_j) - \psi_{\boldsymbol{\theta}^*}(\mathbf{x}_j)\right\| + L_\ell R_{\mathcal{W}} R_\psi \sqrt{\frac{2\log(2/\delta)}{n}}$$

$$\le \; L_\ell L_\psi R_{\mathcal{W}}\left\|\widehat{\boldsymbol{\theta}}_N - \boldsymbol{\theta}^*\right\| + L_\ell R_{\mathcal{W}} R_\psi \sqrt{\frac{2\log(2/\delta)}{n}},$$

where we used Hoeffding's inequality as the third inequality, and the local stability and parameter transfer learnability in the last inequality. Second, we have the following inequality with probability $1 - \bar{\delta}$:

$$
\begin{aligned}
&\mathbb{E}_{(\mathbf{x},y)\sim P}\left[\ell(y, \langle \widehat{\mathbf{w}}_{N,n}, \psi_{\boldsymbol{\theta}^*}(\mathbf{x})\rangle)\right] - \mathbb{E}_{(\mathbf{x},y)\sim P}\left[\ell(y, \langle \widehat{\mathbf{w}}_n^*, \psi_{\boldsymbol{\theta}^*}(\mathbf{x})\rangle)\right] \\
\leq\;\; & L_\ell \mathbb{E}_{(\mathbf{x},y)\sim P}\left[|\langle \widehat{\mathbf{w}}_{N,n}, \psi_{\boldsymbol{\theta}^*}(\mathbf{x})\rangle - \langle \widehat{\mathbf{w}}_n^*, \psi_{\boldsymbol{\theta}^*}(\mathbf{x})\rangle|\right] \\
\leq\;\; & L_\ell R_\psi \left\| \widehat{\mathbf{w}}_{N,n} - \widehat{\mathbf{w}}_n^* \right\|_2 \\
\leq\;\; & L_\ell R_\psi \sqrt{\frac{2 L_\ell L_\psi R_{\mathcal{W}}}{\rho}} \left\| \widehat{\boldsymbol{\theta}}_N - \boldsymbol{\theta}^* \right\|,
\end{aligned}
\tag{4}
$$

where the last inequality is derived by the strong convexity of the regularizer $\rho r(\mathbf{w})$ in the Appendix. Third, the following holds by Theorem 1 of Sridharan et al. (2009) with probability $1 - \delta/2$:

$$
\begin{aligned}
&\mathbb{E}_{(\mathbf{x},y)\sim P}\left[\ell(y, \langle \widehat{\mathbf{w}}_n^*, \psi_{\boldsymbol{\theta}^*}(\mathbf{x})\rangle)\right] - \mathbb{E}_{(\mathbf{x},y)\sim P}\left[\ell(y, \langle \mathbf{w}^*, \psi_{\boldsymbol{\theta}^*}(\mathbf{x})\rangle)\right] \\
=\;\; & \mathbb{E}_{(\mathbf{x},y)\sim P}\left[\ell(y, \langle \widehat{\mathbf{w}}_n^*, \psi_{\boldsymbol{\theta}^*}(\mathbf{x})\rangle) + \rho r(\widehat{\mathbf{w}}_n^*)\right] \\
& - \mathbb{E}_{(\mathbf{x},y)\sim P}\left[\ell(y, \langle \mathbf{w}^*, \psi_{\boldsymbol{\theta}^*}(\mathbf{x})\rangle) + \rho r(\mathbf{w}^*)\right] + \rho(r(\mathbf{w}^*) - r(\widehat{\mathbf{w}}_n^*)) \\
\leq\;\; & \left( \frac{8 L_\ell^2 R_\psi^2 (32 + \log(2/\delta))}{\rho n} \right) + \rho R_r.
\end{aligned}
$$

Thus, when $\rho = L_\ell R_\psi \sqrt{\frac{8(32 + \log(2/\delta))}{R_r n}}$, we have (2) with probability $1 - (\delta + 2\bar{\delta})$. ∎

# 3 Stability and Learnability in Sparse Coding

In this section, we consider the sparse coding in self-taught learning, where the source region essentially consists of the sample space $\mathcal{X}_S$ without the label space $\mathcal{Y}_S$. We assume that the sample spaces in both regions are $\mathbb{R}^d$. Then, the sparse coding method treated here consists of a two-stage procedure, where a dictionary is learnt on the source region, and then a sparse coding with the learnt dictionary is used for a predictive task in the target region.

First, we show that sparse coding satisfies the local stability in Section 3.1 and next explain that appropriate dictionary learning algorithms satisfy the parameter transfer learnability in Section 3.4. As a consequence of Theorem 1, we obtain the learning bound of self-taught learning algorithms based on sparse coding. We note that the results in this section are useful independent of transfer learning.

We here summarize the notations used in this section. Let $\| \cdot \|_p$ be the $p$-norm on $\mathbb{R}^d$. We define as $\mathrm{supp}(\mathbf{a}) := \{i \in [m] | a_i \neq 0\}$ for $\mathbf{a} \in \mathbb{R}^m$. We denote the number of elements of a set $S$ by $|S|$. When a vector $\mathbf{a}$ satisfies $\|\mathbf{a}\|_0 = |\mathrm{supp}(\mathbf{a})| \leq k$, $\mathbf{a}$ is said to be $k$-sparse. We denote the ball with radius $R$ centered at 0 by $B_{\mathbb{R}^d}(R) := \{\mathbf{x} \in \mathbb{R}^d | \|\mathbf{x}\|_2 \leq R\}$. We set as $\mathcal{D} := \{\mathbf{D} = [\mathbf{d}_1, \ldots, \mathbf{d}_m] \in B_{\mathbb{R}^d}(1)^m | \|\mathbf{d}_j\|_2 = 1 \ (i = 1, \ldots, m)\}$ and each $\mathbf{D} \in \mathcal{D}$ a dictionary with size $m$.

**Definition 3** (Induced matrix norm). *For an arbitrary matrix* $\mathbf{E} = [\mathbf{e}_1, \ldots, \mathbf{e}_m] \in \mathbb{R}^{d \times m}$, [1] *the induced matrix norm is defined by* $\|\mathbf{E}\|_{1,2} := \max_{i \in [m]} \|\mathbf{e}_i\|_2$.

We adopt $\| \cdot \|_{1,2}$ to measure the difference of dictionaries since it is typically used in the framework of dictionary learning. We note that $\|\mathbf{D} - \tilde{\mathbf{D}}\|_{1,2} \leq 2$ holds for arbitrary dictionaries $\mathbf{D}, \tilde{\mathbf{D}} \in \mathcal{D}$.

## 3.1 Local Stability of Sparse Representation

We show the local stability of sparse representation under a sparse model. A sparse representation with dictionary parameter $\mathbf{D}$ of a sample $\mathbf{x} \in \mathbb{R}^d$ is expressed as follows:

$$
\varphi_{\mathbf{D}}(\mathbf{x}) := \operatorname*{argmin}_{\mathbf{z} \in \mathbb{R}^m} \frac{1}{2}\|\mathbf{x} - \mathbf{D}\mathbf{z}\|_2^2 + \lambda \|\mathbf{z}\|_1,
$$

where $\lambda > 0$ is a regularization parameter. This situation corresponds to the case where $\boldsymbol{\theta} = \mathbf{D}$ and $\psi_{\boldsymbol{\theta}} = \varphi_{\mathbf{D}}$ in the setting of Section 2.1. We prepare some notions to the stability of the sparse representation. The following margin and incoherence were introduced by Mehta and Gray (2013).

**Definition 4** ($k$-margin). *Given a dictionary* $\mathbf{D} = [\mathbf{d}_1, \ldots, \mathbf{d}_m] \in \mathcal{D}$ *and a point* $\mathbf{x} \in \mathbb{R}^d$, *the $k$-margin of* $\mathbf{D}$ *on* $\mathbf{x}$ *is*

$$\mathcal{M}_k(\mathbf{D}, \mathbf{x}) := \max_{\mathcal{I} \subset [m], |\mathcal{I}| = m-k} \min_{j \in \mathcal{I}} \left\{ \lambda - |\langle \mathbf{d}_j, \mathbf{x} - \mathbf{D}\varphi_{\mathbf{D}}(\mathbf{x}) \rangle| \right\}.$$

**Definition 5** ($\mu$-incoherence). *A dictionary matrix* $\mathbf{D} = [\mathbf{d}_1, \ldots, \mathbf{d}_m] \in \mathcal{D}$ *is termed $\mu$-incoherent if* $|\langle \mathbf{d}_i, \mathbf{d}_j \rangle| \leq \mu/\sqrt{d}$ *for all* $i \neq j$.

Then, the following theorem is obtained.

**Theorem 2** (Sparse Coding Stability). *Let* $\mathbf{D} \in \mathcal{D}$ *be $\mu$-incoherent and* $\|\mathbf{D} - \tilde{\mathbf{D}}\|_{1,2} \leq \lambda$. *When*

$$\|\mathbf{D} - \tilde{\mathbf{D}}\|_{1,2} \leq \epsilon_{k,\mathbf{D}}(\mathbf{x}) := \frac{\mathcal{M}_{k,\mathbf{D}}(\mathbf{x})^2 \lambda}{64 \max\{1, \|\mathbf{x}\|\}^4}, \tag{5}$$

*the following stability bound holds:*

$$\|\varphi_{\mathbf{D}}(\mathbf{x}) - \varphi_{\tilde{\mathbf{D}}}(\mathbf{x})\|_2 \leq \frac{4\|\mathbf{x}\|^2 \sqrt{k}}{(1 - \mu k/\sqrt{d})\lambda} \|\mathbf{D} - \tilde{\mathbf{D}}\|_{1,2}.$$

From Theorem 2, $\epsilon_{k,\mathbf{D}}(\mathbf{x})$ becomes the permissible radius of perturbation in Definition 1.

Here, we refer to the relation with the sparse coding stability (Theorem 4) of Mehta and Gray (2013), who measured the difference of dictionaries by $\| \cdot \|_{2,2}$ instead of $\| \cdot \|_{1,2}$ and the permissible radius of perturbation is given by $\mathcal{M}_{k,\mathbf{D}}(\mathbf{x})^2 \lambda$ except for a constant factor. Applying the simple inequality $\|\mathbf{E}\|_{2,2} \leq \sqrt{m}\|\mathbf{E}\|_{1,2}$ for $\mathbf{E} \in \mathbb{R}^{d \times m}$, we can obtain a variant of the sparse coding stability with the norm $\| \cdot \|_{1,2}$. However, then the dictionary size $m$ affects the permissible radius of perturbation and the stability bound of the sparse coding stability. On the other hand, the factor of $m$ does not appear in Theorem 2, and thus, the result is effective even for a large $m$. In addition, whereas $\|\mathbf{x}\| \leq 1$ is assumed in Mehta and Gray (2013), Theorem 2 does not assume that $\|\mathbf{x}\| \leq 1$ and clarifies the dependency for the norm $\|\mathbf{x}\|$.

In existing studies related to sparse coding, the sparse representation $\varphi_{\mathbf{D}}(\mathbf{x})$ is modified as $\varphi_{\mathbf{D}}(\mathbf{x}) \otimes \mathbf{x}$ (Mairal et al. (2009)) or $\varphi_{\mathbf{D}}(\mathbf{x}) \otimes (\mathbf{x} - \mathbf{D}\varphi_{\mathbf{D}}(\mathbf{x}))$ (Raina et al. (2007)) where $\otimes$ is the tensor product. By the stability of sparse representation (Theorem 2), it can be shown that such modified representations also have local stability.

## 3.2 Sparse Modeling and Margin Bound

In this subsection, we assume a sparse structure for samples $\mathbf{x} \in \mathbb{R}^d$ and specify a lower bound for the $k$-margin used in (5). The result obtained in this section plays an essential role to show the parameter transfer learnability in Section 3.4.

**Assumption 1** (Model). *There exists a dictionary matrix* $\mathbf{D}^*$ *such that every sample* $\mathbf{x}$ *is independently generated by a representation* $\mathbf{a}$ *and noise* $\boldsymbol{\xi}$ *as*

$$\mathbf{x} = \mathbf{D}^* \mathbf{a} + \boldsymbol{\xi}.$$

Moreover, we impose the following three assumptions on the above model.

**Assumption 2** (Dictionary). *The dictionary matrix* $\mathbf{D}^* = [\mathbf{d}_1, \ldots, \mathbf{d}_m] \in \mathcal{D}$ *is $\mu$-incoherent.*

**Assumption 3** (Representation). *The representation* $\mathbf{a}$ *is a random variable that is $k$-sparse (i.e., $\|\mathbf{a}\|_0 \leq k$) and the non-zero entries are lower bounded by $C > 0$ (i.e., $a_i \neq 0$ satisfy $|a_i| \geq C$).*

**Assumption 4** (Noise). *The noise* $\boldsymbol{\xi}$ *is independent across coordinates and sub-Gaussian with parameter $\sigma/\sqrt{d}$ on each component.*

We note that the assumptions do not require the representation $\mathbf{a}$ or noise $\boldsymbol{\xi}$ to be identically distributed while those components are independent. This is essential because samples in the source and target domains cannot be assumed to be identically distributed in transfer learning.

**Theorem 3** (Margin Bound). *Let* $0 < t < 1$. *We set as*

$$\delta_{t,\lambda} \quad := \quad \frac{2\sigma}{(1-t)\sqrt{d}\lambda} \exp\left(-\frac{(1-t)^2 d\lambda^2}{8\sigma^2}\right) + \frac{2\sigma m}{\sqrt{d}\lambda} \exp\left(-\frac{d\lambda^2}{8\sigma^2}\right)$$
$$+ \frac{4\sigma k}{C\sqrt{d(1-\mu k/\sqrt{d})}} \exp\left(-\frac{C^2 d(1-\mu k/\sqrt{d})}{8\sigma^2}\right) + \frac{8\sigma(d-k)}{\sqrt{d}\lambda} \exp\left(-\frac{d\lambda^2}{32\sigma^2}\right). \quad (6)$$

*We suppose that* $d \geq \left\{\left(1 + \frac{6}{(1-t)}\right)\mu k\right\}^2$ *and* $\lambda = d^{-\tau}$ *for arbitrary* $1/4 \leq \tau \leq 1/2$. *Under Assumptions 1-4, the following inequality holds with probability* $1 - \delta_{t,\lambda}$ *at least:*

$$\mathcal{M}_{k,\mathbf{D}^*}(\mathbf{x}) \geq t\lambda. \quad (7)$$

We refer to the regularization parameter $\lambda$. An appropriate reflection of the sparsity of samples requires the regularization parameter $\lambda$ to be set suitably. According to Theorem 4 of Zhao and Yu (2006)[2], when samples follow the sparse model as in Assumptions 1-4 and $\lambda \cong d^{-\tau}$ for $1/4 \leq \tau \leq 1/2$, the representation $\varphi_{\mathbf{D}}(\mathbf{x})$ reconstructs the true sparse representation $\mathbf{a}$ of sample $\mathbf{x}$ with a small error. In particular, when $\tau = 1/4$ (i.e., $\lambda \cong d^{-1/4}$) in Theorem 3, the failure probability $\delta_{t,\lambda} \cong e^{-\sqrt{d}}$ on the margin is guaranteed to become sub-exponentially small with respect to dimension $d$ and is negligible for the high-dimensional case. On the other hand, the typical choice $\tau = 1/2$ (i.e., $\lambda \cong d^{-1/2}$) does not provide a useful result because $\delta_{t,\lambda}$ is not small at all.

### 3.3   Proof of Margin Bound

We give a sketch of proof of Theorem 3. We denote the first term, the second term and the sum of the third and fourth terms of (6) by $\delta_1$, $\delta_2$ and $\delta_3$, respectively From Assumptions 1 and 3, a sample is represented as $\mathbf{x} = \mathbf{D}^*\mathbf{a} + \boldsymbol{\xi}$ and $\|\mathbf{a}\|_0 \leq k$. Without loss of generality, we assume that the first $m - k$ components of $\mathbf{a}$ are 0 and the last $k$ components are not 0. Since

$$\mathcal{M}_{k,\mathbf{D}^*}(\mathbf{x}) \geq \min_{1 \leq j \leq m-k} \lambda - \langle \mathbf{d}_j, \mathbf{x} - \mathbf{D}^* \varphi_{\mathbf{D}}(\mathbf{x}) \rangle = \min_{1 \leq j \leq m-k} \lambda - \langle \mathbf{d}_j, \boldsymbol{\xi} \rangle - \langle \mathbf{D}^{*\top}\mathbf{d}_j, \mathbf{a} - \varphi_{\mathbf{D}}(\mathbf{x}) \rangle,$$

it is enough to show that the following holds an arbitrary $1 \leq j \leq m - k$ to prove Theorem 3:

$$\Pr[\langle \mathbf{d}_j, \boldsymbol{\xi} \rangle + \langle \mathbf{D}^{*\top}\mathbf{d}_j, \mathbf{a} - \varphi_{\mathbf{D}}(\mathbf{x}) \rangle > (1-t)\lambda] \leq \delta_{t,\lambda}. \quad (8)$$

Then, (8) follows from the following inequalities:

$$\Pr\left[\langle \mathbf{d}_j, \boldsymbol{\xi} \rangle > \frac{1-t}{2}\lambda\right] \quad \leq \quad \delta_1, \quad (9)$$

$$\Pr\left[\langle \mathbf{D}^{*\top}\mathbf{d}_j, \mathbf{a} - \varphi_{\mathbf{D}}(\mathbf{x}) \rangle > \frac{1-t}{2}\lambda\right] \quad \leq \quad \delta_2 + \delta_3. \quad (10)$$

The inequality (9) holds since $\|\mathbf{d}_j\| = 1$ by the definition and Assumption 4. Thus, all we have to do is to show (10). We have

$$\langle \mathbf{D}^{*\top}\mathbf{d}_j, \mathbf{a} - \varphi_{\mathbf{D}}(\mathbf{x}) \rangle \quad = \quad \langle [\langle \mathbf{d}_1, \mathbf{d}_j \rangle, \ldots, \langle \mathbf{d}_m, \mathbf{d}_j \rangle]^\top, \mathbf{a} - \varphi_{\mathbf{D}}(\mathbf{x}) \rangle$$
$$= \quad \langle (\mathbf{1}_{\text{supp}(\mathbf{a}-\varphi_{\mathbf{D}}(\mathbf{x}))} \circ [\langle \mathbf{d}_1, \mathbf{d}_j \rangle, \ldots, \langle \mathbf{d}_m, \mathbf{d}_j \rangle])^\top, \mathbf{a} - \varphi_{\mathbf{D}}(\mathbf{x}) \rangle$$
$$\leq \quad \|\mathbf{1}_{\text{supp}(\mathbf{a}-\varphi_{\mathbf{D}}(\mathbf{x}))} \circ [\langle \mathbf{d}_1, \mathbf{d}_j \rangle, \ldots, \langle \mathbf{d}_m, \mathbf{d}_j \rangle]\|_2 \|\mathbf{a} - \varphi_{\mathbf{D}}(\mathbf{x})\|_2, (11)$$

where $\mathbf{u} \circ \mathbf{v}$ is the Hadamard product (i.e. component-wise product) between $\mathbf{u}$ and $\mathbf{v}$, and $\mathbf{1}_A$ for a set $A \subset [m]$ is a vector whose $i$-th component is 1 if $i \in A$ and 0 otherwise.

Applying Theorem 4 of Zhao and Yu (2006) and using the condition for $\lambda$, the following holds with probability $1 - \delta_3$:

$$\text{supp}(\mathbf{a}) = \text{supp}(\varphi_{\mathbf{D}}(\mathbf{x})). \quad (12)$$

Moreover, under (12), the following holds with probability $1 - \delta_2$ by modifying Corollary 1 of Negahban et al. (2009) and using the condition for $\lambda$:

$$\|\mathbf{a} - \varphi_{\mathbf{D}}(\mathbf{x})\|_2 \leq \frac{6\sqrt{k}\lambda}{1 - \frac{\mu k}{\sqrt{d}}}. \tag{13}$$

Thus, if both of (12) and (13) hold, the right hand side of (11) is bounded as follows:

$$\|\mathbf{1}_{\mathrm{supp}(\mathbf{a} - \varphi_{\mathbf{D}}(\mathbf{x}))} \circ [\langle \mathbf{d}_1, \mathbf{d}_j \rangle, \ldots, \langle \mathbf{d}_m, \mathbf{d}_j \rangle]\|_2 \|\mathbf{a} - \varphi_{\mathbf{D}}(\mathbf{x})\|_2$$
$$\leq \quad \sqrt{|\mathrm{supp}(\mathbf{a} - \varphi_{\mathbf{D}}(\mathbf{x}))|} \frac{\mu}{\sqrt{d}} \frac{6\sqrt{k}\lambda}{1 - \frac{\mu k}{\sqrt{d}}} \quad = \frac{6\mu k}{\sqrt{d} - \mu k} \lambda \leq \frac{1-t}{2}\lambda,$$

where we used Assumption 2 in the first inequality, (12) and Assumption 3 in the equality and the condition for $d$ in the last inequality. From the above discussion, the left hand side of (10) is bounded by the sum of the probability $\delta_3$ that (12) does not hold and the probability $\delta_2$ that (12) holds but (13) does not hold. ∎

## 3.4 Transfer Learnability for Dictionary Learning

When the true dictionary $\mathbf{D}^*$ exists as in Assumption 1, we show that the output $\widehat{\mathbf{D}}_N$ of a suitable dictionary learning algorithm from $N$-unlabeled samples satisfies the parameter transfer learnability for the sparse coding $\varphi_{\mathbf{D}}$. Then, Theorem 1 guarantees the learning bound in self-taught learning since the discussion in this section does not assume the label space in the source region. This situation corresponds to the case where $\boldsymbol{\theta}^*_S = \mathbf{D}^*$, $\widehat{\boldsymbol{\theta}}_N = \widehat{\mathbf{D}}_N$ and $\|\cdot\| = \|\cdot\|_{1,2}$ in Section 2.1.

We show that an appropriate dictionary learning algorithm satisfies the parameter transfer learnability for the sparse coding $\varphi_{\mathbf{D}}$ by focusing on the permissible radius of perturbation in (5) under some assumptions. When Assumptions 1-4 hold and $\lambda = d^{-\tau}$ for $1/4 \leq \tau \leq 1/2$, the margin bound (7) for $\mathbf{x} \in \mathcal{X}$ holds with probability $1 - \delta_{t,\lambda}$, and thus, we have

$$\epsilon_{k,\mathbf{D}^*}(\mathbf{x}) \geq \frac{t^2 \lambda^3}{64 \max\{1, \|\mathbf{x}\|\}^4} = \Theta(d^{-3\tau}).$$

Thus, if a dictionary learning algorithm outputs the estimator $\widehat{\mathbf{D}}_N$ such that

$$\|\widehat{\mathbf{D}}_N - \mathbf{D}^*\|_{1,2} \leq \mathcal{O}(d^{-3\tau}) \tag{14}$$

with probability $1 - \delta_N$, the estimator $\widehat{\mathbf{D}}_N$ of $\mathbf{D}^*$ satisfies the parameter transfer learnability for the sparse coding $\varphi_{\mathbf{D}}$ with probability $\bar{\delta} = \delta_N + n\delta_{t,\lambda}$. Then, by the local stability of the sparse representation and the parameter transfer learnability of such a dictionary learning, Theorem 1 guarantees that sparse coding in self-taught learning satisfies the learning bound in (2).

We note that Theorem 1 can apply to any dictionary learning algorithm as long as (14) is satisfied. For example, Arora et al. (2015) show that, when $k = \mathcal{O}(\sqrt{d}/\log d)$, $m = \mathcal{O}(d)$, Assumptions 1-4 and some additional conditions are assumed, their dictionary learning algorithm outputs $\widehat{\mathbf{D}}_N$ which satisfies

$$\|\widehat{\mathbf{D}}_N - \mathbf{D}^*\|_{1,2} = \mathcal{O}(d^{-M})$$

with probability $1 - d^{-M'}$ for arbitrarily large $M, M'$ as long as $N$ is sufficiently large.

## 4 Conclusion

We derived a learning bound (Theorem 1) for a parameter transfer learning problem based on the local stability and parameter transfer learnability, which are newly introduced in this paper. Then, applying it to a sparse coding-based algorithm under a sparse model (Assumptions 1-4), we obtained the first theoretical guarantee of a learning bound in self-taught learning. Although we only consider sparse coding, the framework of parameter transfer learning includes other promising algorithms such as multiple kernel learning and deep neural networks, and thus, our results are expected to be effective to analyze the theoretical performance of these algorithms. Finally, we note that our learning bound can be applied to different settings from self-taught learning because Theorem 1 includes the case in which labeled samples are available in the source region.

## Footnotes

[1] In general, the $(p, q)$-induced norm for $p, q \geq 1$ is defined by $\|\mathbf{E}\|_{p,q} := \sup_{\mathbf{v} \in \mathbb{R}^m, \|\mathbf{v}\|_p = 1} \|\mathbf{E}\mathbf{v}\|_q$. Then, $\| \cdot \|_{1,2}$ in this general definition coincides with that in Definition 3 by Lemma 17 of Vainsencher et al. (2011).

[2] Theorem 4 of Zhao and Yu (2006) is stated for Gaussian noise. However, it can be easily generalized to sub-Gaussian noise as in Assumption 4. Our setting corresponds to the case in which $c_1 = 1/2, c_2 = 1, c_3 = (\log \kappa + \log \log d)/\log d$ for some $\kappa > 1$ (i.e., $e^{d^{c_3}} \cong d^{\kappa}$) and $c_4 = c$ in Theorem 4 of Zhao and Yu (2006). Note that our regularization parameter $\lambda$ corresponds to $\lambda_d/d$ in (Zhao and Yu (2006)).

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
