[Supplementary Material]

# Appendix for **Learning Bound for Parameter Transfer Learning**

## A   Appendix: Lemma for Proof of Theorem 1

In this subsection, we omit the subscript $\mathcal{T}$ for simplicity. In addition, we denote $\boldsymbol{\theta}_{\mathcal{S}}^*$ by $\boldsymbol{\theta}^*$ simply.
We recall

$$
\begin{aligned}
\widehat{\mathbf{w}}_{N,n} &:= \operatorname*{argmin}_{\mathbf{w} \in \mathcal{W}_{\mathcal{T}}} \frac{1}{n} \sum_{j=1}^{n} \ell(y_j, \langle \mathbf{w}, \psi_{\widehat{\boldsymbol{\theta}}_N}(\mathbf{x}) \rangle) + \rho r(\mathbf{w}), \\
\widehat{\mathbf{w}}_n^* &:= \operatorname*{argmin}_{\mathbf{w} \in \mathcal{W}} \frac{1}{n} \sum_{j=1}^{n} \ell(y_j, \langle \mathbf{w}, \psi_{\boldsymbol{\theta}^*}(\mathbf{x}_j) \rangle) + \rho r(\mathbf{w}).
\end{aligned}
$$

The inequality (4) is obtained by the following lemma.

**Lemma 1.** *The following holds with probability $1 - \bar{\delta}$:*

$$
\|\widehat{\mathbf{w}}_{N,n} - \widehat{\mathbf{w}}_n^*\|_2 \leq \sqrt{\frac{2 R_{\mathcal{W}} L_\ell L_\psi}{\rho}} \left\| \widehat{\boldsymbol{\theta}}_N - \boldsymbol{\theta}^* \right\|. \tag{15}
$$

**[Proof]** Let us define as

$$
\begin{aligned}
\widehat{f}_{N,n}(\mathbf{w}) &:= \frac{1}{n} \sum_{j=1}^{n} \ell(y_j, \langle \mathbf{w}, \psi_{\widehat{\boldsymbol{\theta}}_N}(\mathbf{x}) \rangle) + \rho r(\mathbf{w}), \\
\widehat{f}_n^*(\mathbf{w}) &:= \frac{1}{n} \sum_{j=1}^{n} \ell(y_j, \langle \mathbf{w}, \psi_{\boldsymbol{\theta}^*}(\mathbf{x}_j) \rangle) + \rho r(\mathbf{w}).
\end{aligned}
$$

If

$$
\widehat{f}_n^*(\widehat{\mathbf{w}}_n^*) \leq \widehat{f}_{N,n}(\widehat{\mathbf{w}}_{N,n}),
$$

we have the following with probability $1 - \bar{\delta}$:

$$
\begin{aligned}
\widehat{f}_{N,n}(\widehat{\mathbf{w}}_n^*) - \widehat{f}_{N,n}(\widehat{\mathbf{w}}_{N,n}) &\leq \widehat{f}_{N,n}(\widehat{\mathbf{w}}_n^*) - \widehat{f}_n^*(\widehat{\mathbf{w}}_n^*) + \widehat{f}_n^*(\widehat{\mathbf{w}}_n^*) - \widehat{f}_{N,n}(\widehat{\mathbf{w}}_{N,n}) \\
&\leq \widehat{f}_{N,n}(\widehat{\mathbf{w}}_n^*) - \widehat{f}_n^*(\widehat{\mathbf{w}}_n^*) \\
&= \frac{1}{n} \sum_{j=1}^{n} \ell(y_j, \langle \widehat{\mathbf{w}}_n^*, \psi_{\widehat{\boldsymbol{\theta}}_N}(\mathbf{x}_j) \rangle) - \frac{1}{n} \sum_{j=1}^{n} \ell(y_j, \langle \widehat{\mathbf{w}}_n^*, \psi_{\boldsymbol{\theta}^*}(\mathbf{x}_j) \rangle) \\
&\leq \frac{1}{n} \sum_{j=1}^{n} L_\ell \left| \langle \widehat{\mathbf{w}}_n^*, \psi_{\widehat{\boldsymbol{\theta}}_N}(\mathbf{x}_j) \rangle - \langle \widehat{\mathbf{w}}_n^*, \psi_{\boldsymbol{\theta}^*}(\mathbf{x}_j) \rangle \right| \\
&\leq \frac{1}{n} \sum_{j=1}^{n} L_\ell R_{\mathcal{W}} \left\| \psi_{\widehat{\boldsymbol{\theta}}_N}(\mathbf{x}_j) - \psi_{\boldsymbol{\theta}^*}(\mathbf{x}_j) \right\| \\
&\leq \frac{1}{n} \sum_{j=1}^{n} L_\ell R_{\mathcal{W}} L_\psi \left\| \widehat{\boldsymbol{\theta}}_N - \boldsymbol{\theta}^* \right\| \\
&= L_\ell R_{\mathcal{W}} L_\psi \left\| \widehat{\boldsymbol{\theta}}_N - \boldsymbol{\theta}^* \right\|.
\end{aligned}
$$

Since $\widehat{f}_{N,n}$ is $\rho$-strongly convex and $\widehat{\mathbf{w}}_{N,n}$ is its miniizer,

$$
\widehat{f}_{N,n}(\widehat{\mathbf{w}}_n^*) - \widehat{f}_{N,n}(\widehat{\mathbf{w}}_{N,n}) \geq \frac{\rho}{2} \|\widehat{\mathbf{w}}_n^* - \widehat{\mathbf{w}}_{N,n}\|_2^2.
$$

Thus, we obtain (15).

Similarly, if

$$
\widehat{f}_n^*(\widehat{\mathbf{w}}_n^*) \geq \widehat{f}_{N,n}(\widehat{\mathbf{w}}_{N,n}),
$$

we have the following with probability $1 - \bar{\delta}$:

$$
\begin{aligned}
\widehat{f}_n^*(\widehat{\mathbf{w}}_{N,n}) - \widehat{f}_n^*(\widehat{\mathbf{w}}_n^*) &\leq \widehat{f}_n^*(\widehat{\mathbf{w}}_{N,n}) - \widehat{f}_{N,n}(\widehat{\mathbf{w}}_{N,n}) + \widehat{f}_{N,n}(\widehat{\mathbf{w}}_{N,n}) - \widehat{f}_n^*(\widehat{\mathbf{w}}_n^*) \\
&\leq \widehat{f}_n^*(\widehat{\mathbf{w}}_{N,n}) - \widehat{f}_{N,n}(\widehat{\mathbf{w}}_{N,n}) \\
&= \frac{1}{n}\sum_{j=1}^{n}\ell(y_j, \langle \widehat{\mathbf{w}}_{N,n}, \psi_{\boldsymbol{\theta}^*}(\mathbf{x}_j)\rangle) - \frac{1}{n}\sum_{j=1}^{n}\ell(y_j, \langle \widehat{\mathbf{w}}_{N,n}, \psi_{\widehat{\boldsymbol{\theta}}_N}(\mathbf{x}_j)\rangle) \\
&\leq \frac{1}{n}\sum_{j=1}^{n}L_\ell \left| \langle \widehat{\mathbf{w}}_{N,n}, \psi_{\boldsymbol{\theta}^*}(\mathbf{x}_j)\rangle - \langle \widehat{\mathbf{w}}_{N,n}, \psi_{\widehat{\boldsymbol{\theta}}_N}(\mathbf{x}_j)\rangle \right| \\
&\leq \frac{1}{n}\sum_{j=1}^{n}L_\ell R_{\mathcal{W}} \left\| \psi_{\boldsymbol{\theta}^*}(\mathbf{x}_j) - \psi_{\widehat{\boldsymbol{\theta}}_N}(\mathbf{x}_j) \right\| \\
&\leq \frac{1}{n}\sum_{j=1}^{n}L_\ell R_{\mathcal{W}} L_\psi \left\| \widehat{\boldsymbol{\theta}}_N - \boldsymbol{\theta}^* \right\| \\
&= L_\ell R_{\mathcal{W}} L_\psi \left\| \widehat{\boldsymbol{\theta}}_N - \boldsymbol{\theta}^* \right\|.
\end{aligned}
$$

Since $\widehat{f}_n^*$ is $\rho$-strongly convex and $\widehat{\mathbf{w}}_n^*$ is its minimizer,

$$
\widehat{f}_n^*(\widehat{\mathbf{w}}_{N,n}) - \widehat{f}_n^*(\widehat{\mathbf{w}}_n^*) \geq \frac{\rho}{2}\|\widehat{\mathbf{w}}_{N,n} - \widehat{\mathbf{w}}_n^*\|_2^2.
$$

Thus, we obtain (15). ∎

# B   Appendix: Proof of Sparse Coding Stability

The proof of Theorem 2 is almost the same as that of Theorem 1 in Mehta and Gray (2012). However, since a part of the proof can not applied to our setting, we provide the full proof of Theorem 2 in this section.

**Lemma 2.** *Let* $\mathbf{a} \in \mathbb{R}^m$ *and* $\mathbf{E} \in \mathbb{R}^{d \times m}$. *Then,* $\|\mathbf{E}\mathbf{a}\|_2 \leq \|\mathbf{E}\|_{1,2}\|\mathbf{a}\|_1$.

**[Proof]**

$$
\|\mathbf{E}\mathbf{a}\|_2 = \|\sum_{i=1}^{m}a_i\mathbf{e}_i\|_2 \leq \sum_{i=1}^{m}|a_i|\|\mathbf{e}_i\|_2 \leq \|\mathbf{E}\|_{1,2}\sum_{i=1}^{m}|a_i| = \|\mathbf{E}\|_{1,2}\|\mathbf{a}\|_1.
$$

∎

**Lemma 3.** *The sparse representation* $\varphi_{\mathbf{D}}(\mathbf{x})$ *satisfies* $\|\varphi_{\mathbf{D}}(\mathbf{x})\|_1 \leq \frac{\|\mathbf{x}\|_2^2}{2\lambda}$.

**[Proof]**

$$
\begin{aligned}
\lambda\|\varphi_{\mathbf{D}}(\mathbf{x})\|_1 &\leq \frac{1}{2}\|\mathbf{x} - \mathbf{D}\varphi_{\mathbf{D}}(\mathbf{x})\|_2^2 + \lambda\|\varphi_{\mathbf{D}}(\mathbf{x})\|_1 \\
&= \min_{z \in \mathbb{R}^m}\frac{1}{2}\|\mathbf{x} - \mathbf{D}z\|_2^2 + \lambda\|z\|_1 \\
&\leq \frac{1}{2}\|\mathbf{x}\|_2^2.
\end{aligned}
$$

∎

We prepare the following notation:

$$
v_{\mathbf{D}}(\mathbf{z}) := \frac{1}{2}\|\mathbf{x} - \mathbf{D}\mathbf{z}\|_2^2 + \lambda\|\mathbf{z}\|_1.
$$

Let $\mathbf{a}^*$ and $\tilde{\mathbf{a}}^*$ respectively denote the solutions to the LASSO problems for the dictionary $\mathbf{D}$ and $\tilde{\mathbf{D}}$:

$$
\mathbf{a}^* := \operatorname*{argmin}_{\mathbf{z} \in \mathbb{R}^m} v_{\mathbf{D}}(\mathbf{z}), \quad \tilde{\mathbf{a}}^* := \operatorname*{argmin}_{\mathbf{z} \in \mathbb{R}^m} v_{\tilde{\mathbf{D}}}(\mathbf{z})
$$

Then, the following equation holds due to the subgradient of $v_{\mathbf{D}}(\mathbf{z})$ with respect to $\mathbf{z}$ (e.g. (2.8) of **?**).

**Lemma 4.**

$$\lambda\|\mathbf{a}^*\|_1 = \langle \mathbf{x} - \mathbf{D}\mathbf{a}^*, \mathbf{D}\mathbf{a}^* \rangle.$$

Let $v_{\mathbf{D}}$ and $v_{\tilde{\mathbf{D}}}$ be the optimal values of the LASSO problems for the dictionary $\mathbf{D}$ and $\tilde{\mathbf{D}}$:

$$v_{\mathbf{D}} := \min_{\mathbf{z}\in\mathbb{R}^m} v_{\mathbf{D}}(\mathbf{z}) = \frac{1}{2}\|\mathbf{x} - \mathbf{D}\mathbf{a}^*\|_2^2 + \lambda\|\mathbf{a}^*\|_1,$$

$$v_{\tilde{\mathbf{D}}} := \min_{\mathbf{z}\in\mathbb{R}^m} v_{\tilde{\mathbf{D}}}(\mathbf{z}) = \frac{1}{2}\|\mathbf{x} - \tilde{\mathbf{D}}\tilde{\mathbf{a}}^*\|_2^2 + \lambda\|\tilde{\mathbf{a}}^*\|_1.$$

**Lemma 5** (Optimal Value Stability). *If* $\|\mathbf{D} - \tilde{\mathbf{D}}\|_{1,2} \leq \lambda$, *then*

$$|v_{\mathbf{D}} - v_{\tilde{\mathbf{D}}}| \leq \frac{1}{2}\left(1 + \frac{\|\mathbf{x}\|_2}{4}\right)\|\mathbf{x}\|_2^3 \frac{\|\mathbf{D} - \tilde{\mathbf{D}}\|_{1,2}}{\lambda}.$$

**[Proof]**

$$
\begin{aligned}
v_{\tilde{\mathbf{D}}} &\leq \frac{1}{2}\|\mathbf{x} - \tilde{\mathbf{D}}\mathbf{a}^*\|_2^2 + \lambda\|\mathbf{a}^*\|_1 \\
&= \frac{1}{2}\|\mathbf{x} - \mathbf{D}\mathbf{a}^* + (\mathbf{D} - \tilde{\mathbf{D}})\mathbf{a}^*\|_2^2 + \lambda\|\mathbf{a}^*\|_1 \\
&\leq \frac{1}{2}(\|\mathbf{x} - \mathbf{D}\mathbf{a}^*\|_2^2 + 2\|\mathbf{x} - \mathbf{D}\mathbf{a}^*\|_2\|(\mathbf{D} - \tilde{\mathbf{D}})\mathbf{a}^*\|_2 + \|(\mathbf{D} - \tilde{\mathbf{D}})\mathbf{a}^*\|_2^2) + \lambda\|\mathbf{a}^*\|_1 \\
&\leq \frac{1}{2}\|\mathbf{x} - \mathbf{D}\mathbf{a}^*\|_2^2 + \lambda\|\mathbf{a}^*\|_1 + \|\mathbf{x}\|_2\left(\frac{\|\mathbf{x}\|_2^2\|\mathbf{D} - \tilde{\mathbf{D}}\|_{1,2}}{2\lambda}\right) + \frac{1}{2}\left(\frac{\|\mathbf{x}\|_2^2\|\mathbf{D} - \tilde{\mathbf{D}}\|_{1,2}}{2\lambda}\right)^2 \\
&\leq v_{\mathbf{D}} + \left(1 + \frac{\|\mathbf{x}\|_2}{4}\right)\frac{\|\mathbf{x}\|_2^3}{2\lambda}\|\mathbf{D} - \tilde{\mathbf{D}}\|_{1,2},
\end{aligned}
$$

where we used

$$\|\mathbf{x} - \mathbf{D}\mathbf{a}^*\|_2 = \sqrt{\|\mathbf{x} - \mathbf{D}\mathbf{a}^*\|^2} \leq \sqrt{\|\mathbf{x} - \mathbf{D}\mathbf{a}^*\|^2 + \lambda\|\mathbf{a}^*\|_1} \leq \sqrt{\|\mathbf{x}\|_2^2} = \|\mathbf{x}\|_2.$$

∎

The following lemma 6 is obtained by the proof of Lemma 11 in Mehta and Gray (2012).

**Lemma 6** (Stability of Norm of Reconstructor). *If* $\|\mathbf{D} - \tilde{\mathbf{D}}\|_{1,2} \leq \lambda$, *then*

$$\left|\|\mathbf{D}\mathbf{a}^*\|_2^2 - \|\tilde{\mathbf{D}}\tilde{\mathbf{a}}^*\|_2^2\right| \leq 2|v_{\mathbf{D}} - v_{\tilde{\mathbf{D}}}| = \left(1 + \frac{\|\mathbf{x}\|_2}{4}\right)\|\mathbf{x}\|_2^3\frac{\|\mathbf{D} - \tilde{\mathbf{D}}\|_{1,2}}{\lambda}.$$

**Lemma 7.** *If* $\|\mathbf{D} - \tilde{\mathbf{D}}\|_{1,2} \leq \lambda$, *then*

$$\left|\|\mathbf{D}\mathbf{a}^*\|_2^2 - \|\mathbf{D}\tilde{\mathbf{a}}^*\|_2^2\right| \leq (\|\mathbf{x}\|_2 + 3)\|\mathbf{x}\|_2^3\frac{\|\mathbf{D} - \tilde{\mathbf{D}}\|_{1,2}}{\lambda}.$$

**[Proof]** First, note that

$$\|(\tilde{\mathbf{D}} - \mathbf{D})\tilde{\mathbf{a}}^*\|_2 \leq \|(\tilde{\mathbf{D}} - \mathbf{D})\|_{1,2}\|\tilde{\mathbf{a}}^*\|_1 \leq \|\mathbf{x}\|_2^2\frac{\|\mathbf{D} - \tilde{\mathbf{D}}\|_{1,2}}{2\lambda}$$

and

$$
\begin{aligned}
\|\mathbf{D}\tilde{\mathbf{a}}^*\|_2 &\leq \|(\mathbf{D} - \tilde{\mathbf{D}})\tilde{\mathbf{a}}^*\|_2 + \|\tilde{\mathbf{D}}\tilde{\mathbf{a}}^* - \mathbf{x}\|_2 + \|\mathbf{x}\|_2 \\
&\leq \|\mathbf{x}\|_2^2\frac{\|\mathbf{D} - \tilde{\mathbf{D}}\|_{1,2}}{2\lambda} + 2\|\mathbf{x}\|_2 \\
&\leq \left(\frac{1}{2}\|\mathbf{x}\|_2 + 2\right)\|\mathbf{x}\|_2,
\end{aligned}
$$

where we used Lemma 3. Then, we have

$$
\begin{aligned}
& \left| \|\mathbf{D}\tilde{\mathbf{a}}^*\|_2^2 - \|\tilde{\mathbf{D}}\tilde{\mathbf{a}}^*\|_2^2 \right| \\
\leq \quad & 2\left| \langle \mathbf{D}\tilde{\mathbf{a}}^*, (\tilde{\mathbf{D}} - \mathbf{D})\tilde{\mathbf{a}}^* \rangle \right| + \|(\tilde{\mathbf{D}} - \mathbf{D})\tilde{\mathbf{a}}^*\|_2^2 \\
\leq \quad & 2\|\mathbf{D}\tilde{\mathbf{a}}^*\|_2 \|(\tilde{\mathbf{D}} - \mathbf{D})\tilde{\mathbf{a}}^*\|_2 + \|(\tilde{\mathbf{D}} - \mathbf{D})\tilde{\mathbf{a}}^*\|_2^2 \\
\leq \quad & 2\left( \frac{1}{2}\|\mathbf{x}\|_2 + 2 \right) \|\mathbf{x}\|_2 \left( \frac{\|\mathbf{x}\|_2^2 \|\mathbf{D} - \tilde{\mathbf{D}}\|_{1,2}}{2\lambda} \right) + \left( \frac{\|\mathbf{x}\|_2^2 \|\mathbf{D} - \tilde{\mathbf{D}}\|_{1,2}}{2\lambda} \right)^2 \\
\leq \quad & \left( \frac{3}{4}\|\mathbf{x}\|_2 + 2 \right) \|\mathbf{x}\|_2^3 \frac{\|\mathbf{D} - \tilde{\mathbf{D}}\|_{1,2}}{\lambda}.
\end{aligned}
$$

Combining this fact with Lemma 6, we have

$$
\begin{aligned}
& \left| \|\mathbf{D}\mathbf{a}^*\|_2^2 - \|\mathbf{D}\tilde{\mathbf{a}}^*\|_2^2 \right| \\
\leq \quad & \left| \|\mathbf{D}\mathbf{a}^*\|_2^2 - \|\tilde{\mathbf{D}}\tilde{\mathbf{a}}^*\|_2^2 \right| + \left| \|\tilde{\mathbf{D}}\tilde{\mathbf{a}}^*\|_2^2 - \|\mathbf{D}\tilde{\mathbf{a}}^*\|_2^2 \right| \\
\leq \quad & \left( 1 + \frac{\|\mathbf{x}\|_2}{4} \right) \|\mathbf{x}\|_2^3 \frac{\|\mathbf{D} - \tilde{\mathbf{D}}\|_{1,2}}{\lambda} + \left( \frac{3}{4}\|\mathbf{x}\|_2 + 2 \right) \|\mathbf{x}\|_2^3 \frac{\|\mathbf{D} - \tilde{\mathbf{D}}\|_{1,2}}{\lambda} \\
= \quad & \left( \|\mathbf{x}\|_2 + 3 \right) \|\mathbf{x}\|_2^3 \frac{\|\mathbf{D} - \tilde{\mathbf{D}}\|_{1,2}}{\lambda}.
\end{aligned}
$$

∎

**Lemma 8** (Reconstructor Stability). *If $\|\mathbf{D} - \tilde{\mathbf{D}}\|_{1,2} \leq \lambda$, then*

$$
\|\mathbf{D}\mathbf{a}^* - \mathbf{D}\tilde{\mathbf{a}}^*\|_2^2 \leq 2\left( 3\|\mathbf{x}\|_2^2 + 9\|\mathbf{x}\|_2 + 2 \right) \|\mathbf{x}\|_2^2 \frac{\|\mathbf{D} - \tilde{\mathbf{D}}\|_{1,2}}{\lambda}.
$$

**[Proof]** We set as $\bar{\mathbf{a}}^* := \frac{1}{2}(\mathbf{a}^* + \tilde{\mathbf{a}}^*)$. From the optimality of $\mathbf{a}^*$, it follows that $v_{\mathbf{D}}(\mathbf{a}^*) \leq v_{\mathbf{D}}(\bar{\mathbf{a}}^*)$, that is,

$$
\frac{1}{2}\|\mathbf{x} - \mathbf{D}\mathbf{a}^*\|_2^2 + \lambda\|\mathbf{a}^*\|_1 \leq \frac{1}{2}\|\mathbf{x} - \mathbf{D}\bar{\mathbf{a}}^*\|_2^2 + \lambda\|\bar{\mathbf{a}}^*\|_1. \tag{16}
$$

We denote as $\epsilon := \|\mathbf{D} - \tilde{\mathbf{D}}\|_{1,2}$, $c_{\mathbf{x}} := \left( 1 + \frac{\|\mathbf{x}\|_2}{4} \right) \|\mathbf{x}\|_2^3$ and $c'_{\mathbf{x}} := \left( \|\mathbf{x}\|_2 + 3 \right) \|\mathbf{x}\|_2^3$.

By the convexity of the $l_1$-norm, the RHS of (16) obeys:

$$\frac{1}{2}\left\|\mathbf{x}-\mathbf{D}\left(\frac{\mathbf{a}^*+\tilde{\mathbf{a}}^*}{2}\right)\right\|_2^2 + \lambda\left\|\frac{\mathbf{a}^*+\tilde{\mathbf{a}}^*}{2}\right\|_1$$

$$\leq \quad \frac{1}{2}\left\|\mathbf{x}-\frac{1}{2}(\mathbf{D}\mathbf{a}^*+\mathbf{D}\tilde{\mathbf{a}}^*)\right\|_2^2 + \frac{\lambda}{2}\|\mathbf{a}^*\|_1 + \frac{\lambda}{2}\|\tilde{\mathbf{a}}^*\|_1$$

$$= \quad \frac{1}{2}\left(\|\mathbf{x}\|_2^2 - 2\left\langle\mathbf{x},\frac{1}{2}(\mathbf{D}\mathbf{a}^*+\mathbf{D}\tilde{\mathbf{a}}^*)\right\rangle + \frac{1}{4}\|\mathbf{D}\mathbf{a}^*+\mathbf{D}\tilde{\mathbf{a}}^*\|_2^2\right) + \frac{\lambda}{2}\|\mathbf{a}^*\|_1 + \frac{\lambda}{2}\|\tilde{\mathbf{a}}^*\|_1$$

$$= \quad \frac{1}{2}\|\mathbf{x}\|_2^2 - \frac{1}{2}\langle\mathbf{x},\mathbf{D}\mathbf{a}^*\rangle - \frac{1}{2}\langle\mathbf{x},\mathbf{D}\tilde{\mathbf{a}}^*\rangle + \frac{1}{8}(\|\mathbf{D}\mathbf{a}^*\|_2^2 + \|\mathbf{D}\tilde{\mathbf{a}}^*\|_2^2 + 2\langle\mathbf{D}\mathbf{a}^*,\mathbf{D}\tilde{\mathbf{a}}^*\rangle)$$

$$\qquad + \frac{\lambda}{2}\|\mathbf{a}^*\|_1 + \frac{\lambda}{2}\|\tilde{\mathbf{a}}^*\|_1$$

$$\leq \quad \frac{1}{2}\|\mathbf{x}\|_2^2 - \frac{1}{2}\langle\mathbf{x},\mathbf{D}\mathbf{a}^*\rangle - \frac{1}{2}\langle\mathbf{x},\mathbf{D}\tilde{\mathbf{a}}^*\rangle + \frac{1}{4}\|\mathbf{D}\mathbf{a}^*\|_2^2 + \frac{1}{4}\langle\mathbf{D}\mathbf{a}^*,\mathbf{D}\tilde{\mathbf{a}}^*\rangle$$

$$\qquad + \frac{\lambda}{2}\|\mathbf{a}^*\|_1 + \frac{\lambda}{2}\|\tilde{\mathbf{a}}^*\|_1 + \frac{c'_{\mathbf{x}}}{8}\frac{\epsilon}{\lambda}$$

$$= \quad \frac{1}{2}\|\mathbf{x}\|_2^2 - \frac{1}{2}\langle\mathbf{x},\mathbf{D}\mathbf{a}^*\rangle - \frac{1}{2}\langle\mathbf{x},\mathbf{D}\tilde{\mathbf{a}}^*\rangle + \frac{1}{4}\|\mathbf{D}\mathbf{a}^*\|_2^2 + \frac{1}{4}\langle\mathbf{D}\mathbf{a}^*,\mathbf{D}\tilde{\mathbf{a}}^*\rangle$$

$$\qquad + \frac{1}{2}\langle\mathbf{x}-\mathbf{D}\mathbf{a}^*,\mathbf{D}\mathbf{a}^*\rangle + \frac{1}{2}\langle\mathbf{x}-\tilde{\mathbf{D}}\tilde{\mathbf{a}}^*,\tilde{\mathbf{D}}\tilde{\mathbf{a}}^*\rangle + \frac{c'_{\mathbf{x}}}{8}\frac{\epsilon}{\lambda} \qquad (17)$$

$$\leq \quad \frac{1}{2}\|\mathbf{x}\|_2^2 - \frac{1}{2}\langle\mathbf{x},\mathbf{D}\mathbf{a}^*\rangle - \frac{1}{2}\langle\mathbf{x},\mathbf{D}\tilde{\mathbf{a}}^*\rangle + \frac{1}{4}\|\mathbf{D}\mathbf{a}^*\|_2^2 + \frac{1}{4}\langle\mathbf{D}\mathbf{a}^*,\mathbf{D}\tilde{\mathbf{a}}^*\rangle$$

$$\qquad + \frac{1}{2}\langle\mathbf{x},\mathbf{D}\mathbf{a}^*\rangle - \frac{1}{2}\|\mathbf{D}\mathbf{a}^*\|_2^2 + \frac{1}{2}\langle\mathbf{x},\tilde{\mathbf{D}}\tilde{\mathbf{a}}^*\rangle - \frac{1}{2}\|\mathbf{D}\mathbf{a}^*\|_2^2 + \left(\frac{c'_{\mathbf{x}}}{8}+\frac{c_{\mathbf{x}}}{4}\right)\frac{\epsilon}{\lambda}$$

$$= \quad \frac{1}{2}\|\mathbf{x}\|_2^2 - \frac{3}{4}\|\mathbf{D}\mathbf{a}^*\|_2^2 + \frac{1}{4}\langle\mathbf{D}\mathbf{a}^*,\mathbf{D}\tilde{\mathbf{a}}^*\rangle + \frac{1}{2}\langle\mathbf{x},(\tilde{\mathbf{D}}-\mathbf{D})\tilde{\mathbf{a}}^*\rangle + \left(\frac{c'_{\mathbf{x}}+2c_{\mathbf{x}}}{8}\right)\frac{\epsilon}{\lambda},$$

where we used Lemma 4 in (17).

Now, taking the (expanded) LHS of (16) and the newly derived upper bound of the RHS of (16) yields the inequality:

$$\frac{1}{2}\|\mathbf{x}\|_2^2 - \langle\mathbf{x},\mathbf{D}\mathbf{a}^*\rangle + \frac{1}{2}\|\mathbf{D}\mathbf{a}^*\|_2^2 + \lambda\|\mathbf{a}^*\|_1$$

$$\leq \quad \frac{1}{2}\|\mathbf{x}\|_2^2 - \frac{3}{4}\|\mathbf{D}\mathbf{a}^*\|_2^2 + \frac{1}{4}\langle\mathbf{D}\mathbf{a}^*,\mathbf{D}\tilde{\mathbf{a}}^*\rangle + \frac{1}{2}\langle\mathbf{x},(\tilde{\mathbf{D}}-\mathbf{D})\tilde{\mathbf{a}}^*\rangle + \left(\frac{c'_{\mathbf{x}}+2c_{\mathbf{x}}}{8}\right)\frac{\epsilon}{\lambda}.$$

Replacing $\lambda\|\mathbf{a}^*\|_1$ with $\langle\mathbf{x}-\mathbf{D}\mathbf{a}^*,\mathbf{D}\mathbf{a}^*\rangle$ by Lemma 4 yields:

$$-\langle\mathbf{x},\mathbf{D}\mathbf{a}^*\rangle + \frac{1}{2}\|\mathbf{D}\mathbf{a}^*\|_2^2 + \langle\mathbf{x}-\mathbf{D}\mathbf{a}^*,\mathbf{D}\mathbf{a}^*\rangle$$

$$\leq \quad -\frac{3}{4}\|\mathbf{D}\mathbf{a}^*\|_2^2 + \frac{1}{4}\langle\mathbf{D}\mathbf{a}^*,\mathbf{D}\tilde{\mathbf{a}}^*\rangle + \frac{1}{2}\langle\mathbf{x},(\tilde{\mathbf{D}}-\mathbf{D})\tilde{\mathbf{a}}^*\rangle + \left(\frac{c'_{\mathbf{x}}+2c_{\mathbf{x}}}{8}\right)\frac{\epsilon}{\lambda}.$$

Hence,

$$\|\mathbf{D}\mathbf{a}^*\|_2^2 \quad \leq \quad \langle\mathbf{D}\mathbf{a}^*,\mathbf{D}\tilde{\mathbf{a}}^*\rangle + 2\langle\mathbf{x},(\tilde{\mathbf{D}}-\mathbf{D})\tilde{\mathbf{a}}^*\rangle + \left(\frac{c'_{\mathbf{x}}+2c_{\mathbf{x}}}{2}\right)\frac{\epsilon}{\lambda}$$

$$\leq \quad \langle\mathbf{D}\mathbf{a}^*,\mathbf{D}\tilde{\mathbf{a}}^*\rangle + 2\frac{\|\mathbf{x}\|_2^3\epsilon}{2\lambda} + \left(\frac{c'_{\mathbf{x}}+2c_{\mathbf{x}}}{2}\right)\frac{\epsilon}{\lambda}$$

$$= \quad \langle\mathbf{D}\mathbf{a}^*,\mathbf{D}\tilde{\mathbf{a}}^*\rangle + \left(\frac{c'_{\mathbf{x}}+2c_{\mathbf{x}}+2\|\mathbf{x}\|_2^3}{2}\right)\frac{\epsilon}{\lambda}$$

Then, we obtain

$$\|\mathbf{D}\mathbf{a}^* - \mathbf{D}\tilde{\mathbf{a}}^*\|_2^2$$
$$= \|\mathbf{D}\mathbf{a}^*\|_2^2 + \|\mathbf{D}\tilde{\mathbf{a}}^*\|_2^2 - 2\langle \mathbf{D}\mathbf{a}^*, \mathbf{D}\tilde{\mathbf{a}}^*\rangle$$
$$\leq \|\mathbf{D}\mathbf{a}^*\|_2^2 + \left(\|\mathbf{D}\mathbf{a}^*\|_2^2 + c'_{\mathbf{x}}\frac{\epsilon}{\lambda}\right) + \left(-2\|\mathbf{D}\mathbf{a}^*\|_2^2 + (c'_{\mathbf{x}} + 2c_{\mathbf{x}} + 2\|\mathbf{x}\|_2^3)\frac{\epsilon}{\lambda}\right)$$
$$\leq 2(c'_{\mathbf{x}} + c_{\mathbf{x}} + \|\mathbf{x}\|_2^3)\frac{\epsilon}{\lambda}.$$

∎

**Lemma 9.** *[Preservation of Sparsity] If*

$$\mathcal{M}_k(\mathbf{D}, \mathbf{x}) > \left(1 + \frac{\|\mathbf{x}\|_2}{\lambda}\right)\|\mathbf{x}\|_2\|\mathbf{D} - \tilde{\mathbf{D}}\|_{1,2} + \sqrt{2(3\|\mathbf{x}\|_2^2 + 9\|\mathbf{x}\|_2 + 2)\|\mathbf{x}\|_2^2\frac{\|\mathbf{D} - \tilde{\mathbf{D}}\|_{1,2}}{\lambda}}, \quad (18)$$

*then*

$$\|\varphi_{\mathbf{D}}(\mathbf{x}) - \varphi_{\tilde{\mathbf{D}}}(\mathbf{x})\|_0 \leq k. \quad (19)$$

**[Proof]** In this proof, we denote $\varphi_{\mathbf{D}}(\mathbf{x})$ and $\varphi_{\tilde{\mathbf{D}}}(\mathbf{x})$ by $\mathbf{a}^* = [a_1^*, \ldots, a_m^*]^\top$ and $\tilde{\mathbf{a}}^* = [\tilde{a}_1^*, \ldots, \tilde{a}_m^*]^\top$, respectively. When $\tilde{\mathbf{D}} = \mathbf{D}$, Lemma 9 obviously holds. In the following, we assume $\tilde{\mathbf{D}} \neq \mathbf{D}$. Since $\mathcal{M}_k(\mathbf{D}, \mathbf{x}) > 0$ from (18), there is a $\mathcal{I} \subset [m]$ with $|\mathcal{I}| = m - k$ such that for all $i \in \mathcal{I}$:

$$0 < \mathcal{M}_k(\mathbf{D}, \mathbf{x}) \leq \lambda - |\langle \mathbf{d}_j, \mathbf{x} - \mathbf{D}\mathbf{a}^*\rangle|. \quad (20)$$

To obtain (19), it is enough to show that $a_i^* = 0$ and $\tilde{a}_i^* = 0$ for all $i \in \mathcal{I}$.

First, we show $a_i^* = 0$ for all $i \in \mathcal{I}$. From the optimality conditions for the LASSO (**?**), we have

$$\langle \mathbf{d}_j, \mathbf{x} - \mathbf{D}\mathbf{a}^*\rangle = \operatorname{sign}(a_j^*)\lambda \quad \text{if } a_j^* \neq 0,$$
$$|\langle \mathbf{d}_j, \mathbf{x} - \mathbf{D}\mathbf{a}^*\rangle| \leq \lambda \quad \text{otherwise.}$$

Note that the above optimality conditions imply that if $a_j^* \neq 0$ then

$$|\langle \mathbf{d}_j, \mathbf{x} - \mathbf{D}\mathbf{a}^*\rangle| = \lambda. \quad (21)$$

Combining (21) with (20), it holds that $a_i^* = 0$ for all $i \in \mathcal{I}$.

Next, we show $\tilde{a}_i^* = 0$ for all $i \in \mathcal{I}$. To do so, it is sufficient to show that

$$|\langle \tilde{\mathbf{d}}_i, \mathbf{x} - \tilde{\mathbf{D}}\tilde{\mathbf{a}}^*\rangle| < \lambda \quad (22)$$

for all $i \in \mathcal{I}$. Note that

$$|\langle \tilde{\mathbf{d}}_i, \mathbf{x} - \tilde{\mathbf{D}}\tilde{\mathbf{a}}^*\rangle| = |\langle \mathbf{d}_i + \tilde{\mathbf{d}}_i - \mathbf{d}_i, \mathbf{x} - \tilde{\mathbf{D}}\tilde{\mathbf{a}}^*\rangle|$$
$$\leq |\langle \mathbf{d}_i, \mathbf{x} - \tilde{\mathbf{D}}\tilde{\mathbf{a}}^*\rangle| + \|\tilde{\mathbf{d}}_i - \mathbf{d}_i\|_2\|\mathbf{x} - \tilde{\mathbf{D}}\tilde{\mathbf{a}}^*\|_2$$
$$\leq |\langle \mathbf{d}_i, \mathbf{x} - \tilde{\mathbf{D}}\tilde{\mathbf{a}}^*\rangle| + \|\tilde{\mathbf{D}} - \mathbf{D}\|_{1,2}\|\mathbf{x}\|_2$$

and

$$|\langle \mathbf{d}_i, \mathbf{x} - \tilde{\mathbf{D}}\tilde{\mathbf{a}}^*\rangle| = |\langle \mathbf{d}_i, \mathbf{x} - (\mathbf{D} + \tilde{\mathbf{D}} - \mathbf{D})\tilde{\mathbf{a}}^*\rangle|$$
$$\leq |\langle \mathbf{d}_i, \mathbf{x} - \mathbf{D}\tilde{\mathbf{a}}^*\rangle| + |\langle \mathbf{d}_i, (\tilde{\mathbf{D}} - \mathbf{D})\tilde{\mathbf{a}}^*\rangle|$$
$$\leq |\langle \mathbf{d}_i, \mathbf{x} - \mathbf{D}\tilde{\mathbf{a}}^*\rangle| + \|\tilde{\mathbf{D}} - \mathbf{D}\|_{1,2}\|\tilde{\mathbf{a}}^*\|_1.$$

Hence,

$$|\langle \tilde{\mathbf{d}}_i, \mathbf{x} - \tilde{\mathbf{D}}\tilde{\mathbf{a}}^*\rangle| \leq |\langle \mathbf{d}_i, \mathbf{x} - \mathbf{D}\tilde{\mathbf{a}}^*\rangle| + \left(1 + \frac{\|\mathbf{x}\|_2}{\lambda}\right)\|\mathbf{x}\|_2\|\mathbf{D} - \tilde{\mathbf{D}}\|_{1,2}.$$

Now,

$$|\langle \mathbf{d}_i, \mathbf{x} - \mathbf{D}\tilde{\mathbf{a}}^*\rangle| = |\langle \mathbf{d}_i, \mathbf{x} - \mathbf{D}\mathbf{a}^* + \mathbf{D}\mathbf{a}^* - \mathbf{D}\tilde{\mathbf{a}}^*\rangle|$$
$$\leq |\langle \mathbf{d}_i, \mathbf{x} - \mathbf{D}\mathbf{a}^*\rangle| + |\langle \mathbf{d}_i, \mathbf{D}\mathbf{a}^* - \mathbf{D}\tilde{\mathbf{a}}^*\rangle|$$
$$\leq \lambda - \mathcal{M}_k(\mathbf{D}, \mathbf{x}) + \|\mathbf{D}\mathbf{a}^* - \mathbf{D}\tilde{\mathbf{a}}^*\|_2$$
$$\leq \lambda - \mathcal{M}_k(\mathbf{D}, \mathbf{x}) + \sqrt{2(3\|\mathbf{x}\|_2^2 + 9\|\mathbf{x}\|_2 + 2)\|\mathbf{x}\|_2^2\frac{\|\mathbf{D} - \tilde{\mathbf{D}}\|_{1,2}}{\lambda}}, \quad (23)$$

where (23) is due to Lemma 8. Then, (22) is obtained by (18). ∎

Here, we prepare the following lemma.

**Lemma 10.** *When a dictionary $\mathbf{D}$ is $\mu$-incoherent, then the following bound holds for an arbitrary $k$-sparse vector $\mathbf{b}$:*

$$\mathbf{b}^\top \mathbf{D}^\top \mathbf{D} \mathbf{b} \geq \left(1 - \frac{\mu k}{\sqrt{d}}\right) \|\mathbf{b}\|_2^2.$$

**[Proof]** We set as $\mathbf{G} := \mathbf{D}^\top \mathbf{D} - \mathbf{I}$, where $\mathbf{I}$ is the $m \times m$ identity matrix. Since $\mathbf{D}$ is $\mu$-incoherent, the absolute value of each component of $\mathbf{G}$ is less than or equal to $\mu/\sqrt{d}$, and thus, $\mathbf{b}^\top \mathbf{G} \mathbf{b} \geq -\mu/\sqrt{d}\|\mathbf{b}\|_1^2$. Then, we obtain

$$\mathbf{b}^\top \mathbf{D}^\top \mathbf{D} \mathbf{b} \;=\; \mathbf{b}^\top (\mathbf{I} + \mathbf{G})\mathbf{b} \;\geq\; \|\mathbf{b}\|_2^2 - \frac{\mu}{\sqrt{d}}\|\mathbf{b}\|_1^2 \;\geq\; \left(1 - \frac{\mu k}{\sqrt{d}}\right)\|\mathbf{b}\|_2^2, \tag{24}$$

where we used the inequality $\|\mathbf{b}\|_1 \leq \sqrt{k}\|\mathbf{b}\|_2$ for the $k$-sparse vector $\mathbf{b}$ in the last inequality. ∎

**Remark 1.** *We mention the relation with the $k$-incoherence of a dictionary, which is the assumption of the sparse coding stability in Mehta and Gray (2013). For $k \in [m]$ and $\mathbf{D} \in \mathcal{D}$, the $k$-incoherence $s_k(\mathbf{D})$ is defined as*

$$s_k(\mathbf{D}) := (\min\{\varsigma_k(\mathbf{D}_\Lambda)|\Lambda \subset [m], |\Lambda| = k\})^2,$$

*where $\varsigma_k(\mathbf{D}_\Lambda)$ is the $k$-th singular value of $\mathbf{D}_\Lambda = [\mathbf{d}_{i_1}, \ldots, \mathbf{d}_{i_k}]$ for $\Lambda = \{i_1, \ldots, i_k\}$. From Lemma 10, when a dictionary $\mathbf{D}$ is $\mu$-incoherent, the $k$-incoherence of $\mathbf{D}$ satisfies*

$$s_k(\mathbf{D}) \geq 1 - \frac{\mu k}{\sqrt{d}}.$$

*Thus, a $\mu$-incoherent dictionary has positive $k$-incoherence when $d > (\mu k)^2$. On the other hand, when $k \geq 2$, if a dictionary $\mathbf{D}$ has positive $k$-incoherence $s_k(\mathbf{D})$, there is $\mu > 0$ such that the dictionary is $\mu$-incoherent.*

**[Proof of Theorem 2]**

Following by the notations of Mehta and Gray (2012), we denote $\varphi_{\mathbf{D}}(\mathbf{x})$ and $\varphi_{\tilde{\mathbf{D}}}(\mathbf{x})$ by $z_*$ and $t_*$, respectively. From (23) of Mehta and Gray (2012), we have

$$
\begin{aligned}
&(z_* - t_*)^\top \mathbf{D}^\top \mathbf{D}(z_* - t_*) \\
\leq\; & (z_* - t_*)^\top \left((\tilde{\mathbf{D}}^\top \tilde{\mathbf{D}} - \mathbf{D}^\top \mathbf{D})t_* + 2(\mathbf{D} - \tilde{\mathbf{D}})^\top \mathbf{x}\right) \\
=\; & (z_* - t_*)^\top (\tilde{\mathbf{D}}^\top \tilde{\mathbf{D}} - \mathbf{D}^\top \mathbf{D})t_* + 2(z_* - t_*)^\top (\mathbf{D} - \tilde{\mathbf{D}})^\top \mathbf{x}.
\end{aligned}
\tag{25}
$$

We evaluate the second term in (25) [3]. We have the following by the definition of $z_*$:

$$\frac{1}{2}\|\mathbf{x} - \tilde{\mathbf{D}}t_*\|_2^2 + \lambda\|t_*\|_1 \geq \frac{1}{2}\|\mathbf{x} - \tilde{\mathbf{D}}z_*\|_2^2 + \lambda\|z_*\|_1,$$

and thus,

$$2(z_* - t_*)^\top \tilde{\mathbf{D}}^\top \mathbf{x} \geq z_*^\top \tilde{\mathbf{D}}^\top \tilde{\mathbf{D}} z_* - t_*^\top \tilde{\mathbf{D}}^\top \tilde{\mathbf{D}} t_* + 2\lambda(\|z_*\|_1 - \|t_*\|_1).$$

Similarly, we have

$$2(t_* - z_*)^\top \mathbf{D}^\top \mathbf{x} \geq t_*^\top \mathbf{D}^\top \mathbf{D} t_* - z_*^\top \mathbf{D}^\top \mathbf{D} z_* + 2\lambda(\|t_*\|_1 - \|z_*\|_1).$$

Summing up the above inequalities and multiplying $-1$, we obtain

$$
\begin{aligned}
&2(z_* - t_*)^\top (\mathbf{D} - \tilde{\mathbf{D}})^\top \mathbf{x} \\
\leq\; & -z_*^\top \tilde{\mathbf{D}}^\top \tilde{\mathbf{D}} z_* + t_*^\top \tilde{\mathbf{D}}^\top \tilde{\mathbf{D}} t_* - t_*^\top \mathbf{D}^\top \mathbf{D} t_* + z_*^\top \mathbf{D}^\top \mathbf{D} z_* \\
=\; & -z_*^\top (\tilde{\mathbf{D}}^\top \tilde{\mathbf{D}} - \mathbf{D}^\top \mathbf{D}) z_* + t_*^\top (\tilde{\mathbf{D}}^\top \tilde{\mathbf{D}} - \mathbf{D}^\top \mathbf{D}) t_* \\
=\; & (z_* - t_*)^\top (\mathbf{D}^\top \mathbf{D} - \tilde{\mathbf{D}}^\top \tilde{\mathbf{D}}) z_* - (z_* - t_*)^\top (\tilde{\mathbf{D}}^\top \tilde{\mathbf{D}} - \mathbf{D}^\top \mathbf{D}) t_*
\end{aligned}
\tag{26}
$$

$$2(z_* - t_*)^\top (\mathbf{D} - \tilde{\mathbf{D}})^\top \mathbf{x} \;\leq\; 2\|\mathbf{D} - \tilde{\mathbf{D}}\|_{1,2}\sqrt{k}\|z_* - t_*\|_2\|\mathbf{x}\|_2.$$

When $\mathbf{E} := \mathbf{D} - \tilde{\mathbf{D}}$, from (25) and (26),

$$
\begin{aligned}
& (z_* - t_*)^\top \mathbf{D}^\top \mathbf{D}(z_* - t_*) \\
\leq\ & (z_* - t_*)^\top (\mathbf{D}^\top \mathbf{D} - \tilde{\mathbf{D}}^\top \tilde{\mathbf{D}}) z_* \\
\leq\ & |(z_* - t_*)^\top (\mathbf{E}^\top \tilde{\mathbf{D}} + \tilde{\mathbf{D}}^\top \mathbf{E} + \mathbf{E}^\top \mathbf{E}) z_*| \\
\leq\ & |(z_* - t_*)^\top \mathbf{E}^\top \tilde{\mathbf{D}} z_*| + |(z_* - t_*)^\top \tilde{\mathbf{D}}^\top \mathbf{E} z_*| + |(z_* - t_*)^\top \mathbf{E}^\top \mathbf{E} z_*| \\
\leq\ & \|\mathbf{E}(z_* - t_*)\|_2 \|\tilde{\mathbf{D}} z_*\|_2 + \|\tilde{\mathbf{D}}(z_* - t_*)\|_2 \|\mathbf{E} z_*\|_2 + \|\mathbf{E}(z_* - t_*)\|_2 \|\mathbf{E} z_*\|_2 \\
\leq\ & (\|\mathbf{E}\|_{1,2} \|\tilde{\mathbf{D}}\|_{1,2} \|z_*\|_1 + \|\tilde{\mathbf{D}}\|_{1,2} \|\mathbf{E}\|_{1,2} \|z_*\|_1 + \|\mathbf{E}\|_{1,2} \|\mathbf{E}\|_{1,2} \|z_*\|_1) \|z_* - t_*\|_1 \\
\leq\ & \left( \frac{\|\mathbf{x}\|_2^2 \|\mathbf{E}\|_{1,2}}{\lambda} + \frac{\|\mathbf{x}\|_2^2 \|\mathbf{E}\|_{1,2}}{\lambda} + \frac{\|\mathbf{x}\|_2^2 \|\mathbf{E}\|_{1,2}^2}{\lambda} \right) \sqrt{k} \|z_* - t_*\|_2 \\
\leq\ & \left( \frac{4\|\mathbf{x}\|_2^2}{\lambda} \right) \|\mathbf{E}\|_{1,2} \sqrt{k} \|z_* - t_*\|_2,
\end{aligned}
\tag{27}
$$

where we used $\|\mathbf{E}\|_{1,2} \leq 2$ in the last inequality.

We note that the assumption (18) of Lemma 9 follows from (5). Then, since $\|z_* - t_*\|_0 \leq k$ from Lemma 9, we have the following lower bound of (25) from the $\mu$-incoherence of $\mathbf{D}$ and Lemma 10:

$$
(z_* - t_*)^\top \mathbf{D}^\top \mathbf{D}(z_* - t_*) \geq \left( 1 - \frac{\mu k}{\sqrt{d}} \right) \|z_* - t_*\|_2^2.
\tag{28}
$$

By (27) and (28), we obtain

$$
\|z_* - t_*\|_2 \leq \frac{4\|\mathbf{x}\|_2^2 \sqrt{k}}{(1 - \mu k/\sqrt{d})\lambda} \|\mathbf{D} - \tilde{\mathbf{D}}\|_{1,2}.
$$

■

# C   Appendix: Proof of Margin Bound

In this proof, we set as

$$
\begin{aligned}
\delta_1 & := \frac{2\sigma}{(1-t)\sqrt{d}\lambda} \exp\left( -\frac{(1-t)^2 d\lambda^2}{8\sigma^2} \right), \\
\delta_2 & := \frac{2\sigma m}{\sqrt{d}\lambda} \exp\left( -\frac{d\lambda^2}{8\sigma^2} \right), \\
\delta_3' & := \frac{4\sigma k}{C\sqrt{d(1 - \mu k/\sqrt{d})}} \exp\left( -\frac{C^2 d(1 - \mu k/\sqrt{d})}{8\sigma^2} \right) \\
\delta_3'' & := \frac{8\sigma(d-k)}{d\lambda} \exp\left( -\frac{d^2\lambda^2}{32\sigma^2} \right), \\
\delta_3 & := \delta_3' + \delta_3''.
\end{aligned}
$$

Then, $\delta_{t,\lambda} = \delta_1 + \delta_2 + \delta_3$.

The column vectors for a $\mu$-incoherent dictionary are in general position. Thus, a solution of LASSO for a $\mu$-incoherent dictionary is unique due to Lemma 3 in Tibshirani et al. (2013).

The following notions are introduced in Zhao and Yu (2006). Let $\mathbf{a}$ be a $k$-sparse vector. Without loss of generality, we assume that $\mathbf{a} = [a_1, \ldots, a_k, 0, \ldots, 0]^\top$. Then, we denote as $\mathbf{a}(1) = [a_1, \ldots, a_k]^\top$, $\mathbf{D}(1) = [\mathbf{d}_1, \ldots, \mathbf{d}_k]$ and $\mathbf{D}(2) = [\mathbf{d}_{k+1}, \ldots, \mathbf{d}_m]$. Then, we define as $\mathbf{C}_{ij} := \frac{1}{d}\mathbf{D}(i)^\top \mathbf{D}(j)$ for $i, j \in \{1, 2\}$. When a dictionary $\mathbf{D}$ is $\mu$-incoherent and $(\mu k)^2/d < 1$, $\mathbf{C}_{11}$ is positive definite due to Lemma 10 and especially invertible.

**Definition 6** (Strong Irrepresentation Condition). *There exists a positive vector $\boldsymbol{\eta}$ such that*

$$
|\mathbf{C}_{21}\mathbf{C}_{11}^{-1}\mathrm{sign}(\mathbf{a}(1))| \leq \mathbf{1} - \boldsymbol{\eta},
$$

*where $\mathrm{sign}(\mathbf{a}(1))$ maps positive entry of $\mathbf{a}(1)$ to 1, negative entry to $-1$ and 0 to 0, $\mathbf{1}$ is the $(d-k) \times 1$ vector of 1's and the inequality holds element-wise.*

Then, the following lemma is derived by modifying the proof of Corollary 2 of Zhao and Yu (2006).

**Lemma 11** (Strong Irrepresentation Condition)**.** *When a dictionary* $\mathbf{D}$ *is* $\mu$-*incoherent and* $d > \{\mu(2k-1)\}^2$ *holds, the strong irrepresentation condition holds with* $\boldsymbol{\eta} = (1 - \mu(2k-1)/\sqrt{d})\mathbf{1}$.

**Lemma 12.** *Under Assuptions 1-4, when* $\mathbf{D}$ *is* $\mu$-*incoherent and* $d > \mu(2k-1)$, *the following holds:*

$$\Pr\left[|\mathrm{supp}(\mathbf{a} - \varphi_{\mathbf{D}}(\mathbf{x}))| \leq k\right] \geq 1 - \delta_3.$$

**[Proof]** The following inequality obviously holds:

$$\Pr\left[|\mathrm{supp}(\mathbf{a} - \varphi_{\mathbf{D}}(\mathbf{x}))| \leq k\right] \quad \geq \quad \Pr\left[\mathrm{sign}(\mathbf{a}) = \mathrm{sign}(\varphi_{\mathbf{D}}(\mathbf{x}))\right].$$

Due to Lemma 11 and Proofs of Theorems 3 and 4 in Zhao and Yu (2006), there exist sub-Gaussian random variables $\{z_i\}_{i=1}^k$ and $\{\zeta_i\}_{i=1}^{d-k}$ such that their variances are bounded as $\mathbf{E}[z_i^2] \leq \sigma^2/d(1 - \mu k/\sqrt{d}) \leq \sigma^2/d(1 - \mu k/\sqrt{d})$ and $\mathbf{E}[\zeta_i^2] \leq \sigma^2/d^2$ and

$$\Pr\left[\mathrm{sign}(\mathbf{a}) = \mathrm{sign}(\varphi_{\mathbf{D}}(\mathbf{x}))\right]$$

$$\geq \quad 1 - \sum_{i=1}^{k} \Pr\left[|z_i| \geq \sqrt{d}\left(|a_i| - \frac{\sqrt{k}\lambda}{2(1 - \mu k/\sqrt{d})d}\right)\right] - \sum_{i=1}^{d-k} \Pr\left[|\zeta_i| \geq \frac{(1 - \mu(2k-1)/d)\lambda}{2\sqrt{d}}\right].$$

When $\lambda \leq (1 - \mu k/\sqrt{d})Cd/\sqrt{k}$, the inequality $|a_i| - \frac{\sqrt{k}\lambda}{2(1-\mu k/\sqrt{d})d} \geq C/2$ holds since $|a_i| \geq C$. Then, since $1 - \mu(2k-1)/d \geq 1/2$ holds, we obtain

$$\Pr\left[|z_i| \geq \sqrt{d}\left(|a_i| - \frac{\sqrt{k}\lambda}{2(1 - \mu k/\sqrt{d})d}\right)\right] \quad \leq \Pr\left[|z_i| \geq \frac{C\sqrt{d}}{2}\right] \quad \leq \delta_3',$$

$$\Pr\left[|\zeta_i| \geq \frac{(1 - \mu(2k-1)/d)\lambda}{2\sqrt{d}}\right] \quad \leq \Pr\left[|\zeta_i| \geq \frac{\lambda}{4\sqrt{d}}\right] \quad \leq \delta_3'',$$

where we used that $z_i$ and $\zeta_i$ are sub-Gaussian. Thus the proof is completed. ∎

**Lemma 13.** *Let* $\mathbf{D}$ *be a dictionary. When* $\boldsymbol{\xi}$ *satisfies Assumption 4, the following holds:*

$$\Pr[\lambda \geq 2\|\mathbf{D}^\top\boldsymbol{\xi}\|_\infty] \leq 1 - \delta_2,$$

**[Proof]** Let $\xi$ be a 1-dimensional sub-Gaussian with parameter $\sigma/\sqrt{d}$. Then, it holds that for $t > 0$

$$\Pr\left[|\xi| > \lambda\right] \leq \frac{\sigma}{\sqrt{d}\lambda}\exp\left(-\frac{d\lambda^2}{2\sigma^2}\right). \tag{29}$$

Note that $\langle\mathbf{d}_j, \boldsymbol{\xi}\rangle$ is sub-Gaussian with parameter $\sigma/\sqrt{d}$ because $\|\mathbf{d}_j\|_2 = 1$ for every $j \in [m]$ and components of $\boldsymbol{\xi}$ are independent and sub-Gaussian with parameter $\sigma/\sqrt{d}$. Thus,

$$\Pr[\lambda < 2\|\mathbf{D}^\top\boldsymbol{\xi}\|_\infty] = \Pr\left[\cup_{j=1}^m\{\lambda < 2|\langle\mathbf{d}_j, \boldsymbol{\xi}\rangle|\}\right] \leq \sum_{j=1}^m \Pr[\lambda < 2|\langle\mathbf{d}_j, \boldsymbol{\xi}\rangle|] \leq \delta_2,$$

where we used (29) in the last inequality. ∎

**Lemma 14.** *Under Assuptions 1-4, then*

$$\Pr\left[\|\mathbf{a} - \varphi_{\mathbf{D}}(\mathbf{x})\|_2 \leq \frac{3\sqrt{k}}{(1 - \mu k/\sqrt{d})}\lambda\right] \quad \geq \quad 1 - \delta_2 - \delta_3.$$

**[Proof]** By Assumption 1, $\mathbf{x} = \mathbf{D}\mathbf{a} + \boldsymbol{\xi}$. We denote $\varphi_{\mathbf{D}}(\mathbf{x})$ by $\mathbf{a}^*$ and $\mathbf{a} - \mathbf{a}^*$ by $\Delta$. We have the following inequality by the definition of $\mathbf{a}^*$:

$$\frac{1}{2}\|\mathbf{x} - \mathbf{D}\mathbf{a}^*\|_2^2 + \lambda\|\mathbf{a}^*\|_1 \leq \frac{1}{2}\|\mathbf{x} - \mathbf{D}\mathbf{a}\|_2^2 + \lambda\|\mathbf{a}\|_1.$$

Substituting $\mathbf{x} = \mathbf{D}\mathbf{a} + \boldsymbol{\xi}$, we have

$$
\begin{aligned}
\frac{1}{2}\|\mathbf{D}\Delta\|_2^2 &\leq -\langle \mathbf{D}^\top \boldsymbol{\xi}, \Delta \rangle + \lambda(\|\mathbf{a}\|_1 - \|\mathbf{a}^*\|_1) \\
&\leq \|\mathbf{D}^\top \boldsymbol{\xi}\|_\infty \|\Delta\|_1 + \lambda(\|\mathbf{a}\|_1 - \|\mathbf{a}^*\|_1).
\end{aligned}
\tag{30}
$$

Let $\Delta_k$ be the vector whose $i$-th component equals that of $\Delta$ if $i$ is in the support of $\mathbf{a}$ and equals $0$ otherwise. In addition, let $\Delta_k^\perp = \Delta - \Delta_k$. Using $\Delta = \Delta_k + \Delta_k^\perp$, we have

$$
\|\mathbf{a}^*\| = \|\mathbf{a} + \Delta_k^\perp + \Delta_k\|_1 \geq \|\mathbf{a}\|_1 + \|\Delta_k^\perp\|_1 - \|\Delta_k\|_1
$$

Substituting the above inequality into (30), we have

$$
\frac{1}{2}\|\mathbf{D}\Delta\|_2^2 \leq \|\mathbf{D}^\top \boldsymbol{\xi}\|_\infty \|\Delta\|_1 + \lambda(\|\Delta_k\|_1 - \|\Delta_k^\perp\|_1)
$$

The inequality $\lambda \geq 2\|\mathbf{D}^\top \boldsymbol{\xi}\|_\infty$ holds with with probability $1 - \delta_2$ due to Lemma 13, and then, the following inequality holds:

$$
0 \leq \frac{1}{2}\|\mathbf{D}\Delta\|_2^2 \leq \frac{1}{2}\lambda(\|\Delta_k\|_1 + \|\Delta_k^\perp\|_1) + \lambda(\|\Delta_k\|_1 - \|\Delta_k^\perp\|_1).
$$

Thus, $\|\Delta_k^\perp\|_1 \leq 3\|\Delta_k\|_1$ and

$$
\frac{1}{2}\|\mathbf{D}\Delta\|_2^2 \leq \frac{3}{2}\lambda\|\Delta_k\|_1 - \frac{1}{2}\lambda\|\Delta_k^\perp\|_1 \leq \frac{3}{2}\lambda\|\Delta_k\|_1 \leq \frac{3}{2}\lambda\sqrt{k}\|\Delta_k\|_2.
$$

Thus, we have

$$
\|\mathbf{D}\Delta\|_2^2 \leq 3\lambda\sqrt{k}\|\Delta_k\|_2 \leq 3\lambda\sqrt{k}\|\Delta\|_2.
$$

Here, $\|\mathrm{supp}(\Delta)\|_0 \leq k$ with probability $1 - \delta_3$ due to Lemma 12 and the following inequality holds by the $\mu$-incoherence of the dictionary $\mathbf{D}$:

$$
(1 - \mu k/\sqrt{d})\|\Delta\|_2^2 \leq \|\mathbf{D}\Delta\|_2^2,
$$

and thus,

$$
\|\Delta\|_2 \leq \frac{3\lambda\sqrt{k}}{(1 - \mu k/\sqrt{d})}.
$$

$\blacksquare$

[Proof of Theorem 3] From Assumption 1, an arbitrary sample $\mathbf{x}$ is represented as $\mathbf{x} = \mathbf{D}^*\mathbf{a} + \boldsymbol{\xi}$. Then,

$$
\begin{aligned}
\langle \mathbf{d}_j, \mathbf{x} - \mathbf{D}^*\varphi_{\mathbf{D}}(\mathbf{x}) \rangle &= \langle \mathbf{d}_j, \boldsymbol{\xi} + \mathbf{D}^*(\mathbf{a} - \varphi_{\mathbf{D}}(\mathbf{x})) \rangle \\
&= \langle \mathbf{d}_j, \boldsymbol{\xi} \rangle + \langle \mathbf{D}^{*\top}\mathbf{d}_j, \mathbf{a} - \varphi_{\mathbf{D}}(\mathbf{x}) \rangle.
\end{aligned}
$$

Then, we evaluate the probability that the first and second terms is bounded above by $\frac{1-t}{2}\lambda$.

We evaluate the probability for the first term. Since $\|\mathbf{d}_j\| = 1$ by the definition and $\boldsymbol{\xi}$ is drawn from a sub-Gaussian distribution with parameter $\sigma^2/\sqrt{d}$, we have

$$
\Pr\left[ \langle \mathbf{d}_j, \boldsymbol{\xi} \rangle \leq \frac{1-t}{2}\lambda \right] \geq 1 - \delta_1.
$$

With probability $1 - \delta_2 - \delta_3$, the second term is evaluated as follows:

$$
\begin{aligned}
\langle \mathbf{D}^{*\top}\mathbf{d}_j, \mathbf{a} - \varphi_{\mathbf{D}}(\mathbf{x}) \rangle &= \langle [\langle \mathbf{d}_1, \mathbf{d}_j \rangle, \ldots, \langle \mathbf{d}_m, \mathbf{d}_j \rangle]^\top, \mathbf{a} - \varphi_{\mathbf{D}}(\mathbf{x}) \rangle \\
&= \langle (\mathbf{1}_{\mathrm{supp}(\mathbf{a}-\varphi_{\mathbf{D}}(\mathbf{x}))} \circ [\langle \mathbf{d}_1, \mathbf{d}_j \rangle, \ldots, \langle \mathbf{d}_m, \mathbf{d}_j \rangle])^\top, \mathbf{a} - \varphi_{\mathbf{D}}(\mathbf{x}) \rangle \\
&\leq \|(\mathbf{1}_{\mathrm{supp}(\mathbf{a}-\varphi_{\mathbf{D}}(\mathbf{x}))} \circ [\langle \mathbf{d}_1, \mathbf{d}_j \rangle, \ldots, \langle \mathbf{d}_m, \mathbf{d}_j \rangle])^\top\|_2 \|\mathbf{a} - \varphi_{\mathbf{D}}(\mathbf{x})\|_2 \\
&\leq \frac{\mu}{\sqrt{d}}\sqrt{|\mathrm{supp}(\mathbf{a} - \varphi_{\mathbf{D}}(\mathbf{x}))|}\|\mathbf{a} - \varphi_{\mathbf{D}}(\mathbf{x})\|_2 \\
&\leq \frac{3\mu k}{(1 - \mu k/\sqrt{d})\sqrt{d}}\lambda \\
&\leq \frac{1-t}{2}\lambda,
\end{aligned}
$$

$$
\tag{31}
$$
$$
\tag{32}
$$

where we used Lemmas 12 and 14 in (31) and $d \geq \left\{ \left( 1 + \frac{6}{(1-t)} \right) \mu k \right\}^2$ in (32). Thus, with probability $1 - (\delta_1 + \delta_2 + \delta_3) = 1 - \delta_{t,\lambda}$,

$$\mathcal{M}_{k,\mathbf{D}^*}(\mathbf{x}) \geq \lambda - \langle \mathbf{d}_j, \mathbf{x} - \mathbf{D}^* \varphi_{\mathbf{D}}(\mathbf{x}) \rangle \geq t\lambda.$$

Thus, the proof of Theorem 3 is completed. ■

## Footnotes

[3] The following bound in Mehta and Gray (2012) is not used in this paper: