[Reviews · NeurIPS 2016]

Reviewer 1

Summary

The paper studies a problem similar to domain adaptation, where the source domain and target domain slightly differ. The authors consider that the learned model share some of its parameters across both domains, and other parameters are specific to each domain. One can learn the common parameters on the source samples and the target parameters on the target domain. To quantify the feasibility such a task, the authors define a notion of "transfer learnability", and provide a PAC bound on the difference between the risk of a learned model and the best achievable generalization risk. Finally, they apply their bound to sparse coding.

Qualitative Assessment

The parameter transfer learning framework described in the paper is very interesting and deserves attention. The approach taken by the authors (describe in Section 2) is sound but lacks clarity. The notation is well chosen, but not always properly explained (see my "Specific comments" below). Also, as the transfer learning framework is very similar to domain adaptation, which is studied in many papers (as Ben-David et al. 2007 cited by the authors), it would be interesting to discuss the connection of Theorem 1 with existing domain adaptation results. Section 3 is difficult to follow for a reader not familiar with sparse coding (like myself). It would be important to clearly state the protocol involved. I assume that it is a two-stage procedure, which consists of (1) learning a dictionary on the source samples, and then (2) learning a representation on the target samples. This needs to be stated explicitly. Moreover, the authors refer to "self-taught learning" (in the abstract, introduction and conclusion), but never explain explicitly what it is and where it is used in the studied scenario. Finally, the paper contains many sketches or ideas of proofs (Lines 153, 184, 223, footnotes 3 and 4). I enjoin the authors to provide the full proofs as supplementary material. Specific comments: - line 100: \mathcal{W} is never explicitly defined - Definition 1: Please define \theta' and L_\psi. I also think that this definition deserves a reference to other prior work(s) - line 123: [n] is defined at Line 164 - Equation (8): Please define || . ||_{1,2} (This notation seems specialized to dictionaries as described in Section 3). - Moreover, for the reader benefit, Theorem 1 statement should recall the meaning of various constants, as they are scattered in the text. I also think that their influence should be discussed with more emphasis, as Theorem 1 is the main result of the paper. Minor typos: - line 45: self taught (no hyphen) - line 69: It (lower case) - Subsections 2.3 (Analysis of Learning Bound) and 3.3 (Analysis of Margin Bound) should be renamed "Proof of Learning/Margin Bound", because it is what they are. - Equation (28): the line ends with an extra comma.

Confidence in this Review

1-Less confident (might not have understood significant parts)


Reviewer 2

Summary

The paper studies an estimation problem in transfer learning and applies its result to self-taught learning. Given source and target tasks and corresponding hypothesis spaces which share a set \Theta of parameters one wishes to use for the target task the optimal (risk minimizing) parameters for the source task in combination with correspondingly optimal remaining parameters for the target task. The risk associated with this particular combined hypothesis is the base-line in the estimation problem. Since the two distributions are unknown, these optimizations are only approximately possible and one uses the parameters obtained from source data by some fixed algorithm in combination with the output of a regularized algorithm run on the target data to determine the remaining data. The paper provides a bound on the excess risk of this hypothesis to the base-line. The two inter-dependent estimations make this a rather tricky problem. The bound in the paper requires a bound on the estimation error for the source task, a stability property of the feature-maps parametrized by the common parameters, and a strongly convex regularizer in the determination of the residual parameters on the target task. The paper then applies this result to self-taught learning, where the output of sparse coding is used as input for some supervised learning problem.

Qualitative Assessment

I found the results very interesting, both the general result on transfer learning and the application to self taught learning, which gives several novel and interesting aspects with respect to sparse coding. The proofs appear correct. The paper suffers a bit from a hasty last-minute presentation (see comments below), but it certainly merits publication. The authors rebuttal and the discussions have convinced me that lack of clarity is the principal drawback of this paper. I would recommend to throughly revise the paper, if necessary provide additional clarification in an appendix. L114 The shorthand \theta^* isn't introduced until L146. The meaning of "on \theta^*" is unclear. L134 The assumption of strong convexity of the regularizer is absolutely essential in the proof and should be emphasized in the statement of Theorem 1. (8) The subscipts "1,2" on the norms appearantly foreshadow the later application to sparse coding. Here they are uncalled for and only confuse the reader, as \Theta is assumed to be a general Banach space. Again the shorthand \theta^* is used before it is introduced. (17) It would be nice to have a small appendix (supplementary material) with a demonstration of this inequality. L153 A statement of this Theorem would be helpful. L181 Unfortunately no proof is provided.

Confidence in this Review

2-Confident (read it all; understood it all reasonably well)


Reviewer 3

Summary

This paper shows an excess risk bound in a transfer learning setting, with the main emphasis to be the scenario where the data have a latent sparse structure (as in sparse coding). The basic bound, Theorem 1, is proved using an error decomposition into three parts, each of which can be bounded using by-now standard techniques. The main work is establishing (under various assumptions) that a sparse-coding based method learns (from unlabeled ``source'' data) a dictionary that is sufficiently close to the optimal dictionary, and moreover, that under sufficiency proximity, that the sparse codes are stable under dictionary perturbations. The latter result, Theorem 2, was already proved by Mehta and Gray, but the authors offer a variation on this earlier result. They further show (and this constitutes the main technical work of the paper) that under certain assumptions, a notion they call the $k$-margin is large enough to apply the sparse coding stability result.

Qualitative Assessment

\documentclass{article} \usepackage[in]{fullpage} \usepackage{amsmath} \usepackage{amssymb} \usepackage{amsfonts} \begin{document} This paper has an interesting direction and synthesizes a number of results from different previous analyses of sparse coding methods to develop a excess risk bounds for a two-layer (sparse coding and then linear prediction) predictor. While I think this very much is a worthwhile pursuit, I have some concerns with the final (rather implicit) version of Theorem 1 after plugging in the various results from Section 3 (which is concerned with sparse coding, stability, and dictionary learning). I also have some technical concerns that I'll describe shortly. If there are good responses to these concerns, I will increase the score for technical quality (I thought the techniques used in this paper and those it uses from others are highly non-trivial, but the technical score is low right now due to potential correctness issues). Overall, I found Theorem 1 a straightforward result (assuming a sparse coding stability result and the right error decomposition, the proof is not difficult). I did find the analysis in Section 3 to be techniquely involved and to carefully draw from a number of previous results in this space in order to control the dictionary estimation error and the margin. I did not check the proof of Theorem 3 carefully. However, it is concerning that the failur probability in Theorem 3 does not appear to decay with either $N$ or $n$, but only with the dimension $d$. Practically speaking, this may not be an issue, but theoretically, it limits the power of the results as the confidence is fixed with respect to $N$ and $n$. A similar issue occurs elsewhere, but potentially affecting the rate, as I now describe. The most pressing issue is how to interpret Theorem 1 under the setting considered in Section 3 (Assumptions 1 through 4). For Theorem 1 to give a non-trivial bound, we must have $\|\hat{\theta}_N - \theta^*\|_{1,2} = o(n^{-1/2})$ due to the last term of (8). Therefore, looking at (43), in the case that $\tau = \frac{1}{4}$, we need $d = \Omega(n^{2/5})$ (and for $\tau = \frac{1}{2}$ we need $d = \Omega(n^{1/5})$). But $d$ is the ambient dimension, not a free parameter! Therefore, we really do not have the ability to play with $d$ in the bound (this did not seem like a nonparametric setting to me). Therefore, let us turn our attention to the earlier appearing equation (41). If (41) holds as stated, then what prevents us from taking $M, M' \rightarrow \infty$ and thereby concluding that as long as $N$ is polynomial in $d$, then, for all $\varepsilon > 0$, with probability 1 we have $\|\hat{D}_N - D^*\|_{1,2} < \varepsilon$. This feels quite suspect, and so I looked at Theorems 3 and 4 of Arora et al.; their statements say the rate is $d^{-\omega(1)}$,, which I take to mean we have $d^{-h(N)}$ where $\lim_{N \rightarrow \infty} h(N) \rightarrow \infty$. Therefore, to make Theorem 1 useful (say, with a rate of $o(1)$), you need to ensure that $d^{-h(N)} = o(n^{-1/2})$; here I assume that you have made some choice of $N$ to depend on $n$, and typically $N = \Omega(n)$ since unlabeled source data is ``cheap''. Now, for $d^{-h(N)} = o(n^{-1/2})$ to hold seems easy enough if $h(N)$ is linear in $N$, but if it is logarithmic in $N$, one has to be more careful. Therefore, you need to dig into Arora et al.~and present their $\omega(1)$ bound with some additional precision. Please include a response to this in the rebuttal. Regarding Theorem 2, given how different the RHS of (25) is as compared to the corresponding term in Theorem 4 of Mehta and Gray, you should have a proof which at least outlines what changes when doing the analysis under a bound on a $\|\cdot\|_{1,2}$ perturbation instead of their $\|\cdot\|_{2,2} = \|\cdot\|_2$ perturbation. Also, the scaling as $\lambda^3$ is quite disappointing, especially considering that the scaling is only as $\lambda$ in Theorem 4 of Mehta and Gray. Do you think this can be improved, or are perturbations in the $\|\cdot\|_{1,2}$ norm really that much more perilous? It would be helpful to provide a short proof of (17), because in your proof of Theorem 1, it seems that wherever you wrote $\|\hat{\theta}_N - \theta^*\|$, you actually meant to write $\|\hat{\theta}_N - \theta^*\|_{1,2}$, since the latter is what actually appears in the theorem statement. But in (17), it is not clear to me if you should be able to get dependence merely on $\|\hat{\theta}_N - \theta^*\|_{1,2}$ rather than the (typically larger) $\|\hat{\theta}_N - \theta^*\|_{2,2}$. Please clarify this in the rebuttal. Finally, please have your paper checked for English grammatical issues. \subsection*{Minor points} \begin{enumerate} \item I think (using Hoeffding's inequality, a martingale version not being needed here) the last term in (13) should be $L_\ell R_\mathcal{W} R_\psi \sqrt{\frac{2 \log(2/\delta)}{n}}$, since each term is in $[0, 2 R_\psi]$. Also, I don't see how you get $\log(4/\delta)$ since your confidence setting is $\delta/2$. It isn't clear to me that these discrepancies all result from using Azuma-Hoeffding rather than (non-martingale) Hoeffding. \item Assumption 4 is unclear. Do you mean to say that $\xi$ is zero-mean sub-Gaussian with support contained in the ball in $\mathbb{R}^d$ of radius $\sigma / \sqrt{d}$? That is, the case of bounded noise (which is hence sub-Gaussian)? \item On line 256, do you mean to set $M = 5 \tau$ (rather than $M = 5 c$)? \end{enumerate} \subsection*{Update after authors' rebuttal} Regarding my issue with Theorem 3, which is that the failure probability $\delta_{t,,\lambda}$ does not decrease with $n$ or $N$: The authors' response makes sense to me and I agree that this failure probability should be independent of $n$ and $N$. Also, I think in the high-dimensional setting this is acceptable due to the failure probability's exponential decay with $d$, practically speaking. With regards to my issue with the application of Theorem 1 using the sparse coding application of Section 3: Because the authors mentioned that they were not quite correct in some of the additional statements they made beyond (41), but that (41) itself is correct, I had another look at Arora et al., and was frustrated with the imprecision of that paper in terms of using the phrase "high probability" without explaining technically what they actually mean; my conclusion is that I do not know whether or not the authors are correctly interpreting and applying the results of Arora et al., but I am willing to give the authors the benefit of the doubt here. HOWEVER: I want the authors to be very precise on the exact result they are using from Arora et al. (more precise, it seems, then the exposition, of Arora et al. themselves). Moreover, I had thought there may have been an issue with the authors own tuning of their parameters in Section 3, because the application of Theorem 1 in the case of sparse coding appears to give a trivial bound for different settings of $\tau$ unless we have the following limitations (I consider only the cases of $\tau = 1/4$ and $\tau = 1/2$): \begin{itemize} \item If $\tau = 1/4$, then we only apply the bound in the regime $n = O(d^{5/2})$. This seems acceptable, since $n$ is in the target domain, and we often do not assume all that much data in the target domain. \item If $\tau = 1/2$, then we only apply the bound in the regime $n = O(d^5)$. This seems even less of a restriction. \end{itemize} Still, I would like the authors to update their paper to be up-front about these (currently implicit) restrictions. So, my technical concerns are adequately addressed, owing to some optimism on the part of the authors' use of the results of Arora et al. \end{document}

Confidence in this Review

3-Expert (read the paper in detail, know the area, quite certain of my opinion)


Reviewer 4

Summary

This paper formalizes parameter transfer learning, introduces new notions of learn-ability based on local stability and gives a very general sample complexity bound for this formulation. Further, authors apply this framework to sparse coding in self-taught learning problem.

Qualitative Assessment

Parameter transfer has been widely used in transfer learning community. However, no formal theoretical analysis has been conducted to this setting. This paper is the first attempt to formalize this approach. The notions of local stability and transfer learn-ability are intuitive and appropriate for this setting. Theorem 1 shows the error comes from 3 sources. The first term only depends on upper bound for w. Thus showing an improvement over non-transfer learning. The third term goes with n makes sense because estimation error in the source domain is possibly propagate to target domain parameter estimation. Though I think the dependency on n may be tightened. Overall I like this paper to be accepted.

Confidence in this Review

2-Confident (read it all; understood it all reasonably well)


Reviewer 5

Summary

This paper introduces a notion of learnability in a parametric transfer learning context and derives a learning bound for parameter transfer algorithms under local stability assumptions on parametric feature mappings. As an application it considers the performance of sparse coding in self-taught learning. This provides a first theoretical analysis for the parameter transfer setting of self-taught learning considered by Raina et al. In particular, the authors give a theoretical learning bound for self-taught learning under a sparse model.

Qualitative Assessment

* It is essential that you have your paper proofread for English style and grammar issues, e.g. "it would be significant to develop methods which is able to incorporate samples drawn from other distributions.", "abundant samples in other domain", "the high predictive performance stem" and many more such errors, often of mismatched plural/singular, or missing articles for example "aspects of parameter transfer approach", or grammar in general "any learning bound has not been obtained". Many typos are also present, e.g. "has intensively studied encouraged by..." is missing the word "been", erroneous capitalization in "However, It" line 69. * In the initial definition of self-taught learning in line 27 (or the more thorough description in Section 1.1) a key aspect that defines this kind of learning - and which presumably you want to emphasize - is that unlabeled data are not assumed to have a distribution given by a joint (generative) model, though they should in some sense be indicative of structure which will later help in predicting labels, e.g. random images will teach a self-taught learner about basic visual patterns of images, knowledge which can then give an advantage for learning various predictive tasks each with its own joint model. Explicit mention of this aspect of self-taught learning should be made, perhaps rather as an instance of the general principle for unsupervised learning that you do mention in lines 85-87. Another key aspect in the original paper of Rain et al is the parametric aspect - Rain et al introduced the term "self-taught learning" in relation to a coding approach. These bits of the story need to be clarified so the reader can place your work into context. * On line 100 you say "source region" - do you mean "source domain" or are you somehow referring to "part" of the parameter space? * Transfer learning plays a key role in your paper, yet the reference you give for this is Pan and Yang's survey from 2010 with no mention of Baxter. This should be fixed. Since you claim to give "the formulation and terminology in transfer learning" you should cite Baxter 2000 who gave the first theoretical formulation of transfer learning. If for the purposes of your paper you need a parametric re-formulation which tends more towards Pan and Yang you can say so and then cite them too (c.f. line 90). In particular in this "parameter transfer learning" where you drop hypotheses on probability distributions and "learning environment" it would be good to comment this explicitly. Also, say more clearly what you mean by sharing "part of the parameter space", ideally, it would be good to tell the reader what this means in a small learning environment consisting of just source and target. * As a general comment, you should tie in your results with those of Baxter. It is not true there has been no analysis for transfer learning, just that you are using local stability of the parameter transfer map to provide a novel analysis in the parametric setting you consider. In Baxter's case, source tasks were obtained under some probability distribution. In your case that distribution is concentrated on a single source task, but covering numbers and so on can be described in terms of the "sharing" of parameter space in your paper. The reader needs to know what you add to the classic (e.g. Baxter) results for your special case (parametric, small environment, probability concentrated on source task). Your use of stability of the parameter transfer map seems to be one of the main tweaks. * Is Def 1 in 2.2 your own or first given elsewhere? In Def 2 it is misleading to use the very general term "transfer learnability" when you are giving your own definition for the specific setting you consider. Perhaps "parameter transfer learnability"? * The core technical parts of the paper, Section 2.3 and Section 3 look like solid and novel contributions but the initial sections of the paper do not place them properly into context and there is no discussion afterwards of relation to existing theoretical analysis, e.g. of Baxter's in the more general transfer learning setting. With more credible placing into context your technical work would be more broadly accessible.

Confidence in this Review

2-Confident (read it all; understood it all reasonably well)


Reviewer 6

Summary

This paper derives a risk bound for parameter transfer learning and uses the result to give a margin bound for self-taught learning algorithms based on sparse coding.

Qualitative Assessment

1. The problem setting is intrinsically the same as the paper 'Transfer bounds for linear feature learning' by Andreas Maurer. Please compare the risk bound result with the risk bound in Adreas Maurer's paper. 2. The assumption on transfer learnability is very strong, because it not only depends on the stability of the feature mapping, but also depends on the estimator $\hat{\theta}_N$. The assumption says that there exists an estimator $\hat{\theta}_N$ that is close enough to the optimal estimator for population risk minimization. The existence of such an estimator is not trivial.

Confidence in this Review

2-Confident (read it all; understood it all reasonably well)